# FlexiFlow: Decomposable Flow Matching
# for Generation of Flexible Molecular Ensemble

**Riccardo Tedoldi** [1 2]  **Ola Engkvist** [1 3]  **Patrick Bryant** [4 5]  **Hossein Azizpour** [2 5]  **Jon Paul Janet** [1]  **Alessandro Tibo** [1]

## Abstract

Sampling useful three-dimensional molecular structures along with their most favorable conformations is a key challenge in drug discovery. Current state-of-the-art 3D de-novo molecular design generative models are limited to generating a single conformation. However, the conformational landscape of a molecule determines its observable properties and how tightly it is able to bind to a given protein target. By generating a representative set of low-energy conformers, we can more directly assess these properties and potentially improve the ability to generate molecules with desired thermodynamic observables. Towards this aim, we propose *FlexiFlow*, a novel architecture that extends flow-matching models, allowing for the joint sampling of molecules along with multiple conformations while preserving both equivariance and permutation invariance. We demonstrate the effectiveness of our approach on the QM9 and GEOM Drugs datasets, achieving state-of-the-art results in 3D molecular generation producing valid, unique, and novel molecules with high fidelity to the training data distribution. Moreover, we show that our model can generate unstrained conformational ensembles capturing the conformational diversity and providing similar coverage to state-of-the-art physics-based methods at a fraction of the inference time. Finally, FlexiFlow can be successfully transferred to the protein-conditioned ligand generation task, even when the dataset contains only static pockets without accompanying conformations.

---

[1]Molecular AI, Discovery Sciences, R&D AstraZeneca, Gothenburg, Sweden [2]KTH Royal Institute of Technology, Stockholm, Sweden [3]Computer Science and Engineering, Chalmers University of Technology and University of Gothenburg, Sweden [4]Stockholm University, Stockholm, Sweden [5]Science for Life Laboratory, Stockholm, Sweden. Correspondence to: Riccardo Tedoldi <riccardo.tedoldi@gmail.com>.

## 1. Introduction

Flow matching (Lipman et al., 2023) and diffusion models (Ho et al., 2020) now deliver state-of-the-art generation across images, audio, and 3D shapes (Yang et al., 2024), enabled by strong theory and flexible density modeling. Recent work sharpens flow matching for higher fidelity (Domingo-Enrich et al., 2025), compositional generation (Skreta et al., 2025), and faster, more stable, or guided sampling (Liu et al., 2025). We introduce a conditional flow decomposition framework that enables the joint generation of molecular graphs and multiple conformers. Diffusion and flow-matching models have advanced de novo molecular generation (Hoogeboom et al., 2022; Schneuing et al., 2024), protein design (Watson et al., 2023; Anand & Achim, 2022), and protein structure prediction (Jumper et al., 2021; Ingraham et al., 2019), leveraging E(3)/SE(3)-equivariant architectures to capture 3D symmetries. Their application to unconditional 3D small-molecule generation is highly promising (Vignac et al., 2023; Le et al., 2024; Irwin et al., 2025), reaching benchmark ceilings on QM9 (Ramakrishnan et al., 2014) and GEOM Drugs (Axelrod & Gómez-Bombarelli, 2022). However, current models typically produce only a single conformer per molecule, limiting conformational diversity critical for drug discovery, as different conformations can exhibit significantly different biological activities and properties. For a specific target protein, sampling multiple ligand conformations to match distinct interaction patterns is crucial for optimizing binding affinity and selectivity of a drug candidate (Leach et al., 2009).

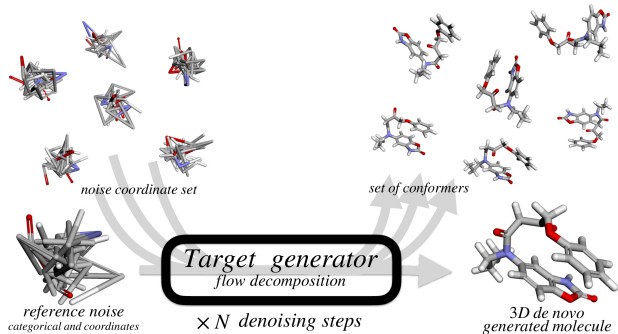

*Figure 1.* From noise samples, our model generates both molecular graphs and their conformational ensembles.

To address this limitation of current models, we propose a novel framework that extends flow matching to jointly generate molecular graphs and representative sets of conformations. To this end, we introduce FlexiFlow [1], a novel architecture that uses this paradigm to handle two sets of coordinates. FlexiFlow preserves permutation invariance on categorical and equivariance on two sets of coordinate features; moreover, it leverages this decomposition to design novel molecular structures along with their 3D conformers (illustration in Figure 1). Our framework shows promising results in generating valid, unique, and novel molecular structures with realistic conformations. Unlike other approaches, such as Boltzmann generators (Noé et al., 2019; Diez et al., 2025), FlexiFlow efficiently generates high-quality molecular structures and conformations while not requiring re-training on additional MD data to produce accurate energy estimations for novel compounds.

We summarize the key contributions as follows:

- We propose a novel framework leveraging conditional independence to decompose the flow matching objective, enabling simultaneous generation of a graph with a single reference conformation along with an arbitrary set of conformers.
- We introduce a new architecture, FlexiFlow, that handles equivariance on two coordinate sets for 3D molecular generation.
- We conduct extensive experiments on benchmark datasets, demonstrating that FlexiFlow achieves state-of-the-art performance in 3D molecular generation tasks.
- We compare FlexiFlow generated conformers with those generated by traditional physics-based methods (CREST (Pracht et al., 2020; 2024)), showing that FlexiFlow generates high-quality conformers at a fraction of the computational cost.
- We show that the FlexiFlow framework transfers effectively to tasks that lack conformation datasets. In particular, we investigate protein conditioning: given a specific static pocket, we generate molecular graphs, each with multiple conformations. This demonstrates the potential of our method for real drug-discovery applications.

## 2. Related Works

Flow matching unifies score-based diffusion and ODE-based generative modeling by learning vector fields that map simple priors to complex data distributions (Lipman et al., 2024). Researchers have improved its scalability (Wildberger et al., 2023) and sample quality (Gat et al., 2024).

---

[1]https://github.com/r1cc4r2o/FlexiFlow

Recent tridimensional denovo drug design methods generate a molecular graph with a single conformer and fall into two categories: (1) generating coordinates first and inferring bonds afterward (Hoogeboom et al., 2022), or (2) jointly generating the graph and bonds (Irwin et al., 2025). EDM (Hoogeboom et al., 2022) addresses this challenge using a diffusion model for 3D molecular structure generation. Other methods (GCDM (Morehead & Cheng, 2024), GFMDiff (Xu et al., 2024), EquiFM (Song et al., 2023), GeomLDM (Xu et al., 2023), GeomBFM (Song et al., 2024), MUDiff (Hua et al., 2023)) often produced unstable structures, but newer architectures and training strategies (FlowMol (Dunn & Koes, 2024), MiDi (Vignac et al., 2023), EQGAT-diff (Le et al., 2024)) have rapidly improved. Conditional flow matching (CFM) is now a leading approach (Song et al., 2023; Campbell et al., 2024), with recent work improving efficiency and generation quality Tabasco (Vonessen et al., 2025), FlowMol3 (Dunn & Koes, 2025), and SemlaFlow (Irwin et al., 2025).

Generating multiple conformations is critical when conditioning on a protein binding pocket (Peng et al., 2022; Dong et al., 2024), as they strongly influence molecular bioactivity and physicochemical properties. Traditional conformer generation relies on physics-based tools such as CREST (Pracht et al., 2024), which search for low-energy conformational minima but are computationally expensive. More recent approaches, such as Adjoint Sampling (Havens et al., 2025), generate conformers using flow matching while keeping the molecular graph fixed, substantially reducing the computational cost of producing conformations that closely approximate minimum-energy states. Building on these advances, we propose a novel conditional flow matching framework that decomposes the flow via a conditional independence assumption, enabling the joint generation of molecular graphs and their corresponding conformations.

## 3. Background

Flow matching offers a versatile and efficient framework to learn a generative process that maps an easy-to-sample distribution $p_{noise}$ to a complex one, often denoted as $p_{data} = q$. To transport $p_{noise}$ to $q$, we can define a marginal probability path $p_t(x)$, parameterized by $t$, that connects the two. For the sake of simplicity, we assume $x \in \mathbb{R}^n$ but the flow matching framework can be extended to more complex structured data. At the beginning of the path, when $t = 0$, $p_0(x) = p_{noise}(x)$, whereas at the end, when $t = 1$, $p_1(x) = q(x)$. The vector field is learned by a neural network $v_t(x; \theta)$ that acts as a mediator between the two distributions and learns to approximate the true vector field $u_t(x)$. The training objective is to minimize the following flow matching loss: $\mathcal{L}_{\text{FM}}(\theta) = \mathbb{E}_{t, x \sim p_t(x)} \|v_t(x; \theta) - u_t(x)\|^2$. In practice, evaluating $p_t(x)$ and $u_t(x)$ is computationally

intractable, as they require integrating over the entire data distribution, which cannot be done analytically. The solution for this problem was proposed by Conditional Flow Matching (CFM), by (Lipman et al., 2023):

$$\mathcal{L}_{\text{CFM}}(\theta) = \mathbb{E}_{\substack{t \sim \mathcal{U}[0,1] \\ x_1 \sim q(x_1) \\ x \sim p_t(x|x_1)}} \left\| v_t(x; \theta) - u_t(x \mid x_1) \right\|^2, \quad (1)$$

where $q(x_1)$ represents the target data distribution, $p_t(x \mid x_1)$ the conditional probability path connecting the prior at $t = 0$ to a distribution concentrated around $x_1$ at $t = 1$, and $u_t(x \mid x_1)$ the target conditional vector field.

## 4. Method

We aim to extend the flow matching paradigm to handle simultaneous flow integration on a set of vectors $\mathcal{S} = \{y_i\}_{i=1}^m$ with a representative vector $x \in \mathcal{S}$.

### 4.1. Flow Decomposition

We consider a time-dependent probability density function $p_t : \mathbb{R}^n \times \mathbb{R}^{n \times m} \to \mathbb{R}$ and $v_t$ a time-dependent vector field $v_t : \mathbb{R}^n \times \mathbb{R}^{n \times m} \to \mathbb{R}^n \times \mathbb{R}^{n \times m}$, where $t \in [0, 1]$. Under the conditional independence assumption on $\mathcal{S} = \{y_i\}_{i=1}^m$, and $p_t$ can be defined as:

$$p_t(x, \mathcal{S}) := \prod_{y \in \mathcal{S}} p_t(x, y). \quad (2)$$

The vector field $v_t$ generates a flow $\psi_t : \mathbb{R}^n \times \mathbb{R}^{n \times m} \to \mathbb{R}^n \times \mathbb{R}^{n \times m}$, which is obtained by solving the corresponding ordinary differential equation (ODE):

$$\frac{\partial \psi_t(x, \mathcal{S})}{\partial t} = v_t(\psi_t(x, \mathcal{S})), \quad \psi_0(x, \mathcal{S}) = (x, \mathcal{S}). \quad (3)$$

We define the flow $\psi_t$ on $(x, \mathcal{S})$ as the concatenation of $m$ independent flows, each of them evaluated on the pair $(x, y)$, $y \in \mathcal{S}$

$$\psi_t(x, \mathcal{S}) = \Big\|_{y \in \mathcal{S}} \psi_t(x, y). \quad (4)$$

The push-forward equation allows us to transform a known distribution (i.e., Gaussian or uniform) $p_0$ into a complex data distribution $p_1$.

$$p_0(x, \mathcal{S}) = p_t(\psi_t(x, \mathcal{S})) \cdot \left[ \det \left( \nabla_{(x, \mathcal{S})} \psi_t(x, \mathcal{S}) \right) \right]. \quad (5)$$

Using Equations 2 and 4, we can decompose push-forward as:

$$\prod_{y \in \mathcal{S}} p_0(x, y) = \prod_{y \in \mathcal{S}} p_t(\psi_t(x, y)) \cdot \left[ \det \left( \nabla_{(x, y)} \psi_t(x, y) \right) \right]. \quad (6)$$

As the flow decomposition is applied to the concatenated set of independent flows, all the findings of (Lipman et al., 2023) remain valid. For the sake of completeness, we provide in Appendix B.1 a proof that the determinant of a block diagonal matrix is the product of the determinants of the blocks. Since the flow $\psi_t(x, \mathcal{S})$ is defined as the concatenation of independent flows $\psi_t(x, y_i)$, we decompose $\psi_t$ into:

$$\frac{\partial \psi_t(x, \mathcal{S})}{\partial t} = \left( \frac{\partial \psi_t(x, y_1)}{\partial t}, \dots, \frac{\partial \psi_t(x, y_m)}{\partial t} \right) \\ = (v_t(\psi_t(x, y_1)), \dots, v_t(\psi_t(x, y_m))). \quad (7)$$

where each $\psi_t(x, y_i)$ is independent of the other $y_j$ for $j \neq i$. Finally, following (Lipman et al., 2023), we can define the flow matching objective for pairs $(x, \mathcal{S})$ as:

$$\mathcal{L}_{\text{FM}}(\theta) = \mathbb{E}_{t, p_t(x, \mathcal{S})} \left\| v_t(x, \mathcal{S}; \theta) - u_t(x, \mathcal{S}) \right\|^2 \\ = \sum_{y \in \mathcal{S}} \mathbb{E}_{t, p_t(x, y)} \left\| v_t(x, y; \theta) - u_t(x, y) \right\|^2. \quad (8)$$

where $v_t$ is modeled with a neural network parametrized by $\theta$. Since $p_t(x, \mathcal{S})$ and $u_t(x, \mathcal{S})$ do not have a closed form, we cannot optimize directly over those terms. However, we can leverage Conditional Flow Matching (CFM) (Lipman et al., 2023) that we extend to pairs $(x, \mathcal{S})$:

$$\mathcal{L}_{\text{CFM}}(\theta) = \mathbb{E}_{\substack{t \sim \mathcal{U} \\ (x_1, \mathcal{S}_1) \sim q \\ (x, \mathcal{S}) \sim p_t}} \left[ \left\| v_t(x, \mathcal{S}; \theta) - u_t(x, \mathcal{S} \mid x_1, \mathcal{S}_1) \right\|^2 \right]. \quad (9)$$

where $q(x_1, \mathcal{S}_1)$ represents the target data distribution, $p_t(x, \mathcal{S} \mid x_1, \mathcal{S}_1)$ the conditional probability path connecting the prior at $t = 0$ to a distribution concentrated around $(x_1, \mathcal{S}_1)$ at $t = 1$, and $u_t(x, \mathcal{S} \mid x_1, \mathcal{S}_1)$ the target conditional vector field.

### 4.2. Flow Decomposition on Molecular Graphs

Our objective is to define a model that generates novel molecular structures along with their low-energy conformations. We denote by $\mathcal{X}$ the molecular space, whose elements are molecular graphs. A molecular graph with $n$ atoms is defined as $\mathcal{G} = \{\mathcal{V}, \mathcal{E}, \mathcal{S}\}$ where $\mathcal{V} \in \mathbb{N}^{n \times 2}$ are the vertices (atom and charge types), $\mathcal{E} \in \mathbb{N}^{n \times n}$ represent the edges (bonds) and $\mathcal{S} = \{y_1, \dots, y_m \mid y_i \in \mathbb{R}^{n \times 3}\}$ a set of $m$ conformations. There is a one-to-one correspondence between atoms in each conformation and nodes in the graph. We decomposed each molecular graph into a set of 4-tuples

$$\mathcal{D}_{\mathcal{G}} = \{(\mathcal{V}, \mathcal{E}, x, y) \mid y \in \mathcal{S}\}, \quad (10)$$

where $x \in \mathcal{S}$ is the representative conformation. We denote with $\mathcal{D}$ the full dataset, that is the union over all the set of 4-tuples $\mathcal{D}_{\mathcal{G}}$. There are multiple ways to choose $x$ and alternative reference conformer selection strategies

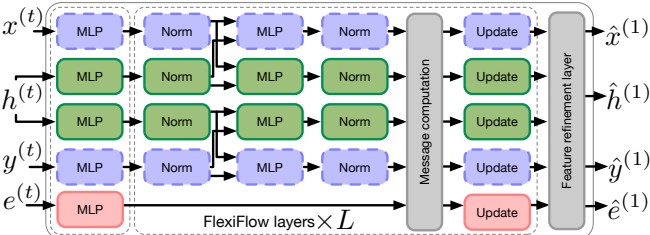

*Figure 2.* The FlexiFlow architecture takes equivariant and invariant features as input at time $t \in [0, 1]$ and produces predictions at $t = 1$. The left macro dashed block is the featurization layer. Blocks in the same column with the same color share weights. Solid blocks represent invariant features, while dashed blocks represent equivariant features. Message computation and feature refinement layer blocks produce both invariant and equivariant features.

(i.e. closest to average, minimum-energy conformer) were considered but yielded indistinguishable results due to negligible differences in the energy state of closest to average and minimum-energy conformer (Appendix E). In our case, we select the conformation that is closest to the average conformation, i.e., $x = \arg\min_{y \in \mathcal{S}} \|y - \frac{1}{N} \sum_{y \in \mathcal{S}} y\|^2$.

**Equivariance.** To support the generation of each data point $(\mathcal{V}, \mathcal{E}, x, y) \in \mathcal{D}$, our architecture supports two separate inputs for $x$ and $y$, and maintains equivariance to rotation not only when the same rotation $R$ is applied to both $x$ and $y$, but also when the two different rotations $R_x \neq R_y$ are applied independently. This allows the model to focus on learning the symmetries while retaining equivariance. In the following section, we will show that by using the scalar product on the feature coordinates (see Equation 13), we can construct messages conditioned on both $x$ and $y$, while preserving equivariance.

### 4.3. FlexiFlow Model

We introduce the FlexiFlow architecture which supports the generation of $(\mathcal{V}, \mathcal{E}, x, y)$, where $x$ and $y$ belong to the set $\mathcal{S}$, and $x$ is the representative conformation. FlexiFlow is inspired by SemlaFlow, originally proposed by (Irwin et al., 2025). In this section, we highlight the differences and novelties introduced by FlexiFlow. The architecture is depicted in Figure 2, defined as the composition of one featurization layer, $L$ repeated FlexiFlow layers, and a final feature refinement layer. The most relevant architectural differences compared to (Irwin et al., 2025) are: (1) FlexiFlow requires an extra input $y$, (2) invariant features are used to condition both $x$ and $y$, (3) FlexiFlow exchanges information between $x$ and $y$, while preserving equivariance on both $x$ and $y$. Full details of the entire architecture are deferred to the Appendix A.1. With minor architectural modification (all detailed in Appendix A.3), we additionally extended FlexiFlow to support protein pocket conditioning ligand generation.

Following the SemlaFlow notation, the atom and charge types in $\mathcal{V}$ and the bond types in $\mathcal{E}$ are represented as one hot vectors, denoted by $h$ and $e$, respectively. Note that $h$ is formed by concatenating two one-hot vectors: one encoding the atom type and the other encoding the charge type. Furthermore, $h$ is concatenated with temporal information $t \in [0, 1]$. For ease of exposition, we adopt a slight abuse of notation, overwriting the symbols $h, e, x, y$ to denote their transformed representations through the *featurization layer*. The featurization layer produces invariant features $h \in \mathbb{R}^{n \times d}$ and edge features $e \in \mathbb{R}^{n \times n \times d}$ with two multilayer perceptrons (MLPs). A shared linear layer maps $x$ and $y$ into coordinate sets $x \in \mathbb{R}^{n \times d \times 3}$ and $y \in \mathbb{R}^{n \times d \times 3}$ (see Appendix A.1 for details). The feature tensors $h, e, x,$ and $y$ are then fed into $L$ stacked *FlexiFlow* layers. Each FlexiFlow layer consists of a feed forward layer followed by a graph attention layer. The feed forward consists of two MLPs $\Phi_\theta$ and $\Psi_\theta$ for invariant and equivariant features respectively. Each invariant features $h_i^x$ and $h_i^y$, where $h_i = h_i^x = h_i^y$ in the first FlexiFlow layer, are updated considering also the equivariant features as follows:

$$h_i^{x\,\mathrm{ff}} = h_i^x + \Phi_\theta([\, \tilde{h}_i^x \,, \|\tilde{x}_i\| \,]) \quad h_i^{y\,\mathrm{ff}} = h_i^y + \Phi_\theta([\, \tilde{h}_i^y \,, \|\tilde{y}_i\| \,]),\tag{11}$$

where $[\cdot, \cdot]$ denotes concatenation and $\tilde{x}_i,$ and $\tilde{y}_i$ are normalized invariant features obtained through normalization layers (see Appendix A.1). Note that the norm of a coordinate set is defined component-wise, i.e., $\|x_i\| = [\|x_{i,1}\|, \ldots, \|x_{i,d}\|]$. This design choice allows the invariant features to propagate information to $x$ and $y$ independently. The equivariant feature update is also performed independently on $x_i$ and $y_i$ as follows:

$$x_i^{\mathrm{ff}} = x_i + W_g \Big( \sum_{j=1}^{d} (W_f \, \tilde{x}_j) \otimes \Psi_\theta(\tilde{h}_i^x) \Big)$$

$$y_i^{\mathrm{ff}} = y_i + W_g \Big( \sum_{j=1}^{d} (W_f \, \tilde{y}_j) \otimes \Psi_\theta(\tilde{h}_i^y) \Big),\tag{12}$$

where $W_f$ and $W_g$ are two linear projections (see Appendix A.1).

These features $x^{\mathrm{ff}}, y^{\mathrm{ff}}, h^{x\,\mathrm{ff}}$ and $h^{y\,\mathrm{ff}}$ are used as input to the graph attention layer in combination with the edge features $e^x$ and $e^y$, where in the first FlexiFlow layer $e = e^x = e^y$. Similar to the feed-forward layer, the graph attention layer shares the same normalization layers and MLPs for both invariant and equivariant features. The key difference from SemlaFlow, however, is that we now combine the features corresponding to $x$ and $y$. The messages are computed as follows:

$$x_p = \tilde{x}_i^{\mathrm{ff}} \cdot \tilde{x}_j^{\mathrm{ff}\,T}, \quad y_p = \tilde{y}_i^{\mathrm{ff}} \cdot \tilde{y}_j^{\mathrm{ff}\,T}, \tag{13}$$

$$h_p^x = [W_h \tilde{h}_i^{x\,\mathrm{ff}} \parallel W_j \tilde{h}_j^{x\,\mathrm{ff}}], \quad h_p^y = [W_h \tilde{h}_i^{y\,\mathrm{ff}} \parallel W_h \tilde{h}_j^{y\,\mathrm{ff}}]$$

$$\omega_p^x = [h_p^x, x_p, e^x], \quad \omega_p^y = [h_p^x \cdot h_p^y, x_p, y_p, e^x \cdot e^y]$$

where $W_h$ represents a linear projection and the final messages are computed using two separate MLPs on $\omega_p^x$ and $\omega_p^y$, as they have different input dimensions. These messages are then used to update $x^{\text{ff}}$, $y^{\text{ff}}$, $h^{x\,\text{ff}}$, $h^{y\,\text{ff}}$, $e^x$ and $e^y$ (see Appendix A.1 for complete details). Note that the scalar product, i.e. when calculating $x_p$, is to be understood component-wise, i.e., $x_i \cdot x_j^T = [x_{i,1} x_{j,1}^T, \ldots x_{i,d} x_{j,d}^T]$. Sharing the information between $x$ and $y$ as described in Equation 13 allows the FlexiFlow model to retain the equivariance on the coordinates.

**Theorem 4.1** (Equivariance). *The FlexiFlow model is equivariant with respect to the coordinates $x$ and $y$.*

*Proof.* Let $\tilde{x}$ and $\tilde{y}$ be the normalized coordinate sets as defined in Equation 13, and let $R_x \in SO(3)$ and $R_y \in SO(3)$ be rotation matrices. The only exchange of information between $x$ and $y$ occurs in Equation 13. Applying any rotations $R_x$ and $R_y$ to $\tilde{x}$ and $\tilde{y}$ does not affect the scalar product, since

$$R_x \tilde{x}_i \tilde{x}_j^T R_x^T = \tilde{x}_i R_x R_x^T \tilde{x}_j^T = \tilde{x}_i \tilde{x}_j^T.$$

The same argument applies to $\tilde{y}$. Since SemlaFlow (Irwin et al., 2025) is equivariant with respect to the coordinates, the remainder of the proof follows directly. $\qquad \square$

The *features refinement layer* applies a feed forward layer on $x^{\text{ff}}$, $y^{\text{ff}}$, $h^{x\,\text{ff}}$, $h^{y\,\text{ff}}$ followed by an edge features aggregator on $e^x$, $e^y$. Lastly, three shared MLPs are used to predict the logits for atoms, charges types from $h^{x\,\text{ff}}$ and bonds types from $e^x$. In Appendix A.1 we provide a detailed mathematical formulation for each component of the architecture.

**Loss.** Following Equation 9, the model is trained to minimize the coupled conditional flow-matching loss, which now additionally incorporates the categorical loss component. We reformulate the loss over molecular graphs as a composition of different terms: $\mathcal{L}_{x,y}$ coordinates loss for $x$ and $y$ computed as the mean squared error between the predicted and target, $\mathcal{L}_a$, $\mathcal{L}_c$, $\mathcal{L}_e$ computed as the negative log likelihood between the predicted and target atoms, charges and bonds types, respectively. We also used a regularization term $\mathcal{L}_{\text{reg}}$ to enforce physically valid bond lengths and to encourage low divergence between the categorical features of $x$ and $y$. The full loss is thus defined as: $\mathcal{L}_{FlexiFlow} = \mathcal{L}_{x,y} + \mathcal{L}_a + \mathcal{L}_c + \mathcal{L}_e + \mathcal{L}_{\text{reg}}$. We provide an extensive description of each loss component as part of the Appendix A.4.

**Inference.** Algorithm 1 describes the inference scheme. Let $\mathcal{A}$, $\mathcal{B}$, and $\mathcal{C}$ denote the sets of atom types, bond types, and charge types, respectively. We denote with $(a, b, c) \sim \text{Cat}(1/|\mathcal{A}|) \cdot \text{Cat}(1/|\mathcal{B}|) \cdot \text{Cat}(1/|\mathcal{C}|)$, the sampling from three independent categorical distributions with uniform probability over each set. We denote with $f_\theta$ a trained

---

**Algorithm 1** Inference scheme

1: **Input:** $\Delta t$
2: $x_t \sim \mathcal{N}(0, \mathrm{I})$, $y_t \sim \mathcal{N}(0, \mathrm{I})$, $t \leftarrow 0$
3: $a_t, b_t, c_t \sim \text{Cat}(1/|\mathcal{A}|) \cdot \text{Cat}(1/|\mathcal{B}|) \cdot \text{Cat}(1/|\mathcal{C}|)$
4: **while** $t < 1$ **do**
5: $\quad (\hat{x}_1, \hat{y}_1, \hat{a}_1, \hat{b}_1, \hat{c}_1) \leftarrow f_\theta(x_t, y_t, a_t, b_t, c_t)$
6: $\quad x_t \leftarrow x_t + \Delta t\,(\hat{x}_1 - x_t)/(1 - t)$
7: $\quad y_t \leftarrow y_t + \Delta t\,(\hat{y}_1 - y_t)/(1 - t)$
8: $\quad a_t \leftarrow \text{CATUPDATE}(\hat{a}_1, a_t, t, \Delta t)$
9: $\quad b_t \leftarrow \text{CATUPDATE}(\hat{b}_1, b_t, t, \Delta t)$
10: $\quad c_t \leftarrow \text{CATUPDATE}(\hat{c}_1, c_t, t, \Delta t)$
11: $\quad t \leftarrow t + \Delta t$
12: **end while**

---

FlexiFlow model parametrized by $\theta$. $\Delta t$ represents the time step. The function CATUPDATE is used to perform the update features at inference time, and applies the strategy developed by (Campbell et al., 2024). Since $x_t$ and $y_t$ are sampled independently, our flow decomposition permits two modes of sampling: (1) drawing both $x_t$ and $y_t$ from $\mathcal{N}(0, \mathrm{I})$, or (2) fixing $x_t$ to a specific noise configuration while sampling $y_t \sim \mathcal{N}(0, \mathrm{I})$. This allows us to generate distinct molecular graphs, each with multiple conformations, with only an additional $\sim 30\%$ inference time overhead to also sample the $y$ conformer (see Appendix F).

## 5. Experiments

In this section, we evaluate FlexiFlow's performance in three-dimensional molecular generation and the quality of the resulting energies. To the best of our knowledge, Flexi-Flow is the first model to jointly generate molecular graphs and multiple conformations. We therefore decompose the evaluation into two parts. First, in Section 5.1, we compare FlexiFlow against single-conformer molecular generative models. Second, in Section 5.2, we compare it against physics-based conformer generation methods, which do not generate molecular graphs but instead evaluate the energy landscape of given molecules. We emphasize that Flexi-Flow addresses a more challenging task than existing state-of-the-art methods; nevertheless, it achieves comparable or superior performance relative to competing approaches. Finally, in Section 5.3, we evaluate FlexiFlow on a protein-conditioned generation task.

To assess the generative capabilities of FlexiFlow, we compare it against several generative models that produce a single conformer per molecule: EDM (Hoogeboom et al., 2022), GCDM (Morehead & Cheng, 2024), GFMD-iff (Xu et al., 2024), EquiFM (Song et al., 2023), GeomLDM (Xu et al., 2023), GeomBFM (Song et al., 2024), MUDiff (Hua et al., 2023), FlowMol (Dunn & Koes, 2024), MiDi (Vignac et al., 2023), EQGAT-diff (Le et al., 2024), Tabasco (Vonessen et al., 2025), FlowMol3 (Dunn & Koes,

2025), and SemlaFlow (Irwin et al., 2025). Following the evaluation protocols used by prior work, we report atomic and molecular stability (i.e., valid electron configurations), validity (compliance with basic chemical rules), novelty (the fraction of generated molecules absent from the training set), and uniqueness (the fraction of distinct molecular graphs).

To assess the quality of the multiple conformers generated for each molecular graph, we evaluate their structural diversity using the following metric:

$$D(S) = \frac{1}{|S|} \sum_{x \in S} \min_{\substack{y \in S \\ y \neq x}} \mathrm{RMSD}(x, y), \qquad (14)$$

which measures conformational diversity after optimal alignment and indicates whether energy minimization causes the conformers to collapse into shared local minima. To further assess the extent to which the generated conformers cover the low-energy space, we employ the computationally expensive state-of-the-art physics-based method CREST (Pracht et al., 2024). In this setting, we report Absolute Mean RMSD (AMR) and Coverage (Cov) with respect to low-energy references (see Appendix A.9). Both metrics are defined in terms of precision (P) and recall (R): AMR-R computes the average RMSD from each CREST-generated conformers to its closest generated conformer, while AMR-P computes the average RMSD from each generated conformer to its closest CREST-generated conformers. To compute the Cov metrics, we use a threshold $\delta = \{0.0, \ldots, 2.5\}$ Å with step $0.125$ Å and report Cov-R and Cov-P. Cov-R($\delta$) is the fraction of CREST-generated conformers that have at least one generated conformer within RMSD $\delta$, Cov-P($\delta$) is the fraction of generated conformers that have at least one CREST-generated conformers within $\delta$.

Finally, to illustrate its potential, we demonstrate the generation of ligands conditioned on PDBBind protein pockets (Wang et al., 2005) and present qualitative MNIST results in Appendix D.

## 5.1. Molecular Generation

**Training set-up.** We use QM9 (Ramakrishnan et al., 2014) and GEOM Drugs (Axelrod & Gómez-Bombarelli, 2022) datasets to train our model using the training split for both from (Vignac et al., 2023; Le et al., 2024; Irwin et al., 2025) training the model with the hydrogens. Since QM9 lacks multiple conformers, 20 RDKit conformers are generated per molecule, while GEOM Drugs already provides on average $23\pm10$ semi-empirical DFT conformers per molecule. See Appendix A.7 for further details on the data processing for both datasets. We trained the model on QM9 for 40 epochs and on GEOM Drugs for 4 epochs, refer to Appendix A.5, for further details on the training.

We compare FlexiFlow with recent state-of-the-art molecular generative models on the QM9 and GEOM Drugs

*Table 1.* The table shows the results on QM9, where the methods are grouped into those which infer bonds from coordinates (top) and those which generate bonds directly (bottom). Methods marked with * publish only results over molecules that are both unique and valid. Tabasco authors reported 34% Novelty (no Uniqueness reported); we achieve 90%. See Table 4. NFE refers to the number of inference steps.

| Model | Atom Stab ↑ | Mol Stab ↑ | Valid ↑ | Unique ↑ | NFE |
|---|---|---|---|---|---|
| EDM | 98.7 | 82.0 | 91.9 | 98.9* | 1000 |
| GCDM | 98.7 | 85.7 | 94.8 | 98.4* | 1000 |
| GFMDiff | 98.9 | 87.7 | 96.3 | 98.8* | 500 |
| EquiFM | 98.9 | 88.3 | 94.7 | 98.7* | 210 |
| GeoLDM | 98.9 | 89.4 | 93.8 | 98.8 | 1000 |
| MUDiff | 98.8 | 89.9 | 95.3 | 99.1 | 1000 |
| GeoBFN | 99.3 | 93.3 | 96.9 | 95.4 | 2000 |
| FlowMol | 99.7 | 96.2 | 97.3 | – | 100 |
| MiDi | 99.8 | 97.5 | 97.9 | 97.6 | 500 |
| Tabasco | – | – | 100.0 | – | 100 |
| EQGAT-diff | $\mathbf{99.9}_{\pm 0.0}$ | $98.7_{\pm 0.18}$ | $99.0_{\pm 0.16}$ | $\mathbf{100.0}_{\pm 0.0}$ | 500 |
| SemlaFlow | $\mathbf{99.9}_{\pm 0.0}$ | $\mathbf{99.7}_{\pm 0.03}$ | $99.4_{\pm 0.03}$ | $95.4_{\pm 0.12}$ | 100 |
| FLEXIFLOW | $\mathbf{100.0}_{\pm 0.0}$ | $\mathbf{99.9}_{\pm 0.01}$ | $\mathbf{99.9}_{\pm 0.01}$ | $\mathbf{100.0}_{\pm 0.00}$ | 100 |

*Table 2.* The table shows the results on GEOM Drugs, where the methods are grouped into those which infer bonds from coordinates (top) and those which generate bonds directly (bottom). Methods marked with * uses the estimates for the molecule stability provided by (Irwin et al., 2025) as the papers do not report this metric. Tabasco results show the best model with guidance. NFE refers to the number of inference steps.

| Model | Atom Stab ↑ | Mol Stab ↑ | Valid ↑ | Unique ↑ | Novel ↑ | NFE |
|---|---|---|---|---|---|---|
| EDM | 81.3 | 0.0* | – | – | – | 1000 |
| GCDM | 89.0 | 5.2 | – | – | – | 1000 |
| MUDiff | 84.0 | 60.9 | 98.9 | – | – | 1000 |
| GFMDiff | 86.5 | 3.9 | – | – | – | 500 |
| EquiFM | 84.1 | 0.0* | 98.9 | – | – | – |
| GeoBFN | 86.2 | 0.0* | 91.7 | – | – | 2000 |
| GeoLDM | 98.9 | 61.5* | $\mathbf{99.3}$ | – | – | 1000 |
| FlowMol | 99.0 | 67.5 | 51.2 | – | – | 100 |
| MiDi | $\mathbf{99.8}$ | 91.6 | 77.8 | $\mathbf{100.0}$ | $\mathbf{100.0}$ | 500 |
| EQGAT-diff | $99.8_{\pm 0.0}$ | $93.4_{\pm 0.21}$ | $94.6_{\pm 0.24}$ | $\mathbf{100.0}_{\pm 0.0}$ | $99.9_{\pm 0.07}$ | 500 |
| Flowmol3 | – | – | $\mathbf{99.9}_{\pm 0.10}$ | – | – | 250 |
| Tabasco | – | – | $\mathbf{97.0}_{\pm 0.10}$ | – | 92.0 | 100 |
| SemlaFlow | $99.8_{\pm 0.0}$ | $97.3_{\pm 0.08}$ | $93.9_{\pm 0.19}$ | $\mathbf{100.0}_{\pm 0.0}$ | $99.6_{\pm 0.03}$ | 100 |
| FLEXIFLOW | $\mathbf{99.9}_{\pm 0.0}$ | $\mathbf{99.9}_{\pm 0.01}$ | $92.0_{\pm 0.10}$ | $\mathbf{100.0}_{\pm 0.0}$ | $99.9_{\pm 0.01}$ | 100 |

datasets. For this comparison, we sample $30k$ molecules from each trained model. As shown in Tables 1 and 2, FlexiFlow achieves state-of-the-art performance on both datasets. It matches competing methods in terms of uniqueness and validity, while outperforming them in novelty, atomic stability, and molecular stability on both GEOM Drugs and QM9. Again, we emphasize that FlexiFlow can generate an arbitrary number of conformations for a given molecule, unlike all other methods.

## 5.2. Energy Evaluation

To evaluate how well the generated conformers explore different energy minima, we generate six sets of 20 molecular

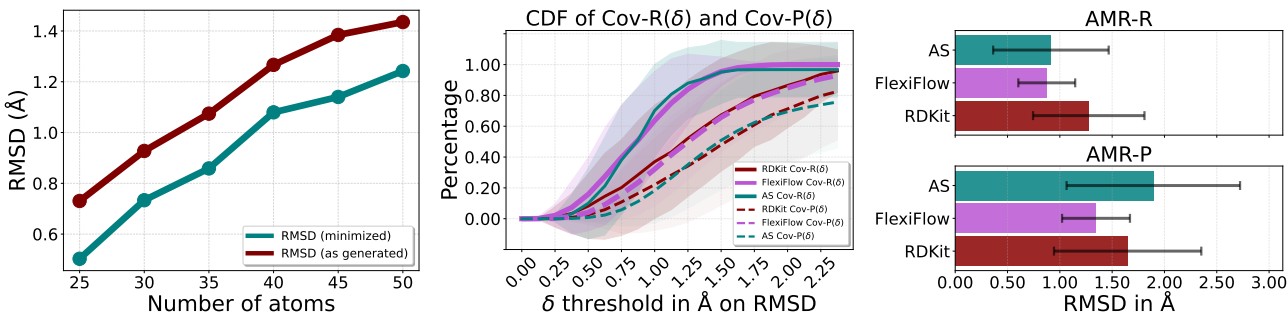

*Figure 3.* RMSD before and after energy minimization (left), and precision–recall curves for Cov (middle) and AMR (right) on the GEOM Drugs dataset, comparing CREST-generated conformers to the ones generated by RDKit, FlexiFlow, and Adjoint Sampling (AS). See Appendix A.9 for Cov and AMR definitions.

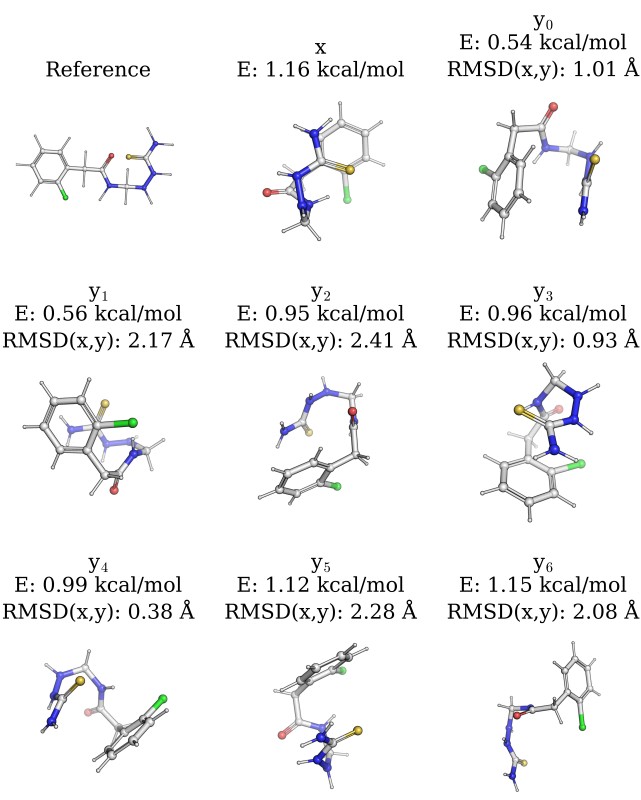

*Figure 4.* The molecular graph of the generated molecule in 2D for illustration purposes (top right). The rest of the grid shows the reference molecule ($x$), followed by a set of generated conformers ($y$). The $y$ conformers are aligned to the $x$ conformer for better visualization and we report energy (E) and the RMSD value between each generated conformer ($y$) and the reference one ($x$).

structures, each containing molecules with 25, 30, 35, 40, 45, and 50 atoms, respectively. For each set $\mathcal{S}$, we compute the diversity metric defined in Equation 14. We also consider the corresponding set $\mathcal{S}^*$, in which all conformers are energy-minimized. Evaluating $\mathcal{S}^*$ provides a stricter test, as minimization can potentially collapse different conformers into the same structure.

We observe (see Figure 3, left) that the minimum pairwise distance between conformers consistently increases with the number of atoms, even after minimization. This indicates that, even in the worst-case scenario the states tend to occupy distinct energy minima rather than collapsing into the same structure (see Appendix A.10 for details).

We benchmark FlexiFlow against RDKit ETKDG, Adjoint Sampling (AS) (Havens et al., 2025), and CREST: for 100 molecules we sample 300 conformers each with FlexiFlow (100 NFE), use the lowest-energy FlexiFlow conformer to seed CREST (Pracht et al., 2024) and ETKDG (Riniker & Landrum, 2015) reference ensembles, and run AS with the same NFE to generate its conformers.

In Figure 3 middle and right, we report Coverage (Cov) and Average Minimum RMSD (AMR) between the generated conformers and the optimal energy minima conformers obtained with CREST. These are reported in terms of recall (R) and precision (P) (see Appendix A.9 for details). Figure 3 (middle) shows the cumulative distributions for Cov-R($\delta$) (solid lines) and Cov-P($\delta$) (dashed lines) for RDKit, Adjoint Sampling (AS), and FlexiFlow. For both Cov-R($\delta$) and Cov-P($\delta$), the earliest the curves reach 1 the better. As shown, FlexiFlow performs at least as well as AS on Cov-R($\delta$), and it outperforms both AS and RDKit on Cov-P($\delta$). Those results suggest that AS tends to produce conformers clustered around fewer energy minima, whereas FlexiFlow explores the conformational space more broadly, covering more distinct minima, as indicated by Cov-P($\delta$). The drop in Cov-P($\delta$) for AS is partly due to the high novelty of the molecular graphs produced by 3D generative models (close to $\sim$100%), which makes generalization to new molecules more challenging. Addressing this for AS would require generating conformers for these systems and re-training or fine-tuning the model which becomes quickly infeasible for each generated molecule. Figure 3 (right) shows AMR-R and AMR-P, where lower values indicate better performance. FlexiFlow outperforms both AS and RDKit on average.

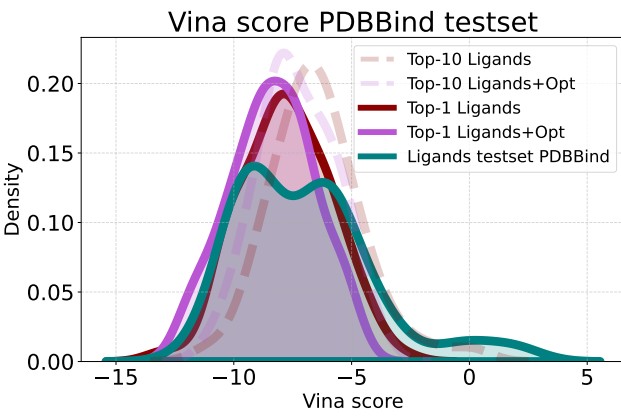

*Figure 5.* Vina score distribution for the generated ligands on PDBBind testset proteins. Opt reports results after slight ligand adjustments using MMFF94[1] conditioned to the protein.

Finally, Figure 4 illustrates an example of a generated molecule where its $y$ conformers are in a different state compared to the $x$ state. We report the energy of the conformers along with the RMSD with respect to the reference state $x$. More detailed results are reported in Appendix C.1, C.2, and C.3.

**Ablation comparing different model settings:** We conduct an ablation study in Appendix C.5, comparing FlexiFlow's performance across model sizes, training epochs, and number of function evaluations (NFE) on QM9 (100 molecules × 300 conformers each, 30k molecules per run). Scaling from small (S) to large (L) models and increasing training epochs both improve performance across all metrics. However, increasing NFE from 100 to 500 yields no significant improvement in strain energies, indicating that the model generates high-quality samples efficiently with fewer inference steps.

### 5.3. Protein Conditioning

We provide some additional experiments on targeted generation with protein conditioning, training FlexiFlow on a subset of 8k protein-ligand complexes training samples from PDBBind (Wang et al., 2004) with ligand QED > 0.5 and tested using the (Corso et al., 2023) splits.

During training, for 300 epochs we interleave GEOM Drugs batches with PDBBind protein–ligand complexes batches to support multiple conformers; since PDBBind lacks multiple ligands per protein, we reuse the same ligand conformation $x$ for each ligand conformation $y$. This approach allows the model to leverage flexibility transfer, where the conformational diversity learned from GEOM Drugs is preserved and applied to the protein-conditioned generation task.

By sampling 120 molecules for each testset protein, our

---

[1]https://github.com/MolecularAI/TorchMMFF94

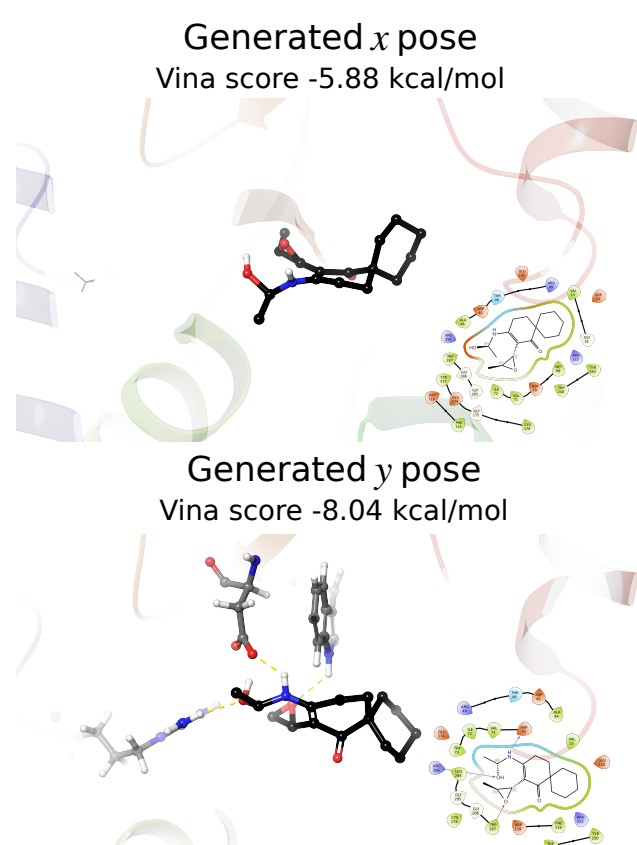

*Figure 6.* Figure reports $x$ and $y$ generated poses for a generated ligand on protein 6jb0 (QED=0.78).

method finds better Vina scores (in kcal/mol) on $y$ conformers in 2,124 cases out of 13,440. Figure 5 shows the Vina score (Eberhardt et al., 2021) distribution of top-1 and top-10 ligands sampled per testset protein across the 13,440 unique $x$–$y$ ligand pairs. FlexiFlow achieves an average Vina score of –7.4 kcal/mol for the top-1 prediction comparable with the target testset data distribution. We emphasize that Vina scores have limited accuracy but offer a practical balance between reproducibility and computational cost, unlike Free Energy Perturbation methods (Wang et al., 2019), which are more accurate but highly resource intensive and sensitive to system setup.

In a similar experimental setting, we trained a model variant that excluded the molecular flexibility information, by omitting the GEOM Drugs dataset during training. As reported in Appendix C.7, the model trained with GEOM Drugs, exhibited improved strained-energy profiles for the generated top-$k$ $y$ ligand conformations. This suggests that incorporating molecular flexibility can enhance conformation generation performance on other downstream task, even when specific conformation datasets are missing.

Figure 6 illustrates a testset example on protein 6jb0, where one of the $y$ conformers achieves a better Vina score than $x$.

# 6. Conclusion

We introduced FlexiFlow, a novel approach that leverages conditional independence to decompose the flow for the simultaneous generation of 3D molecular graphs and conformer sets. The FlexiFlow architecture preserves equivariant properties for coordinates and invariant properties for atom types. We demonstrated its effectiveness in both de novo molecular generation and conformer generation, achieving results comparable to or better than existing methods while producing diverse sets of high-quality conformations. In addition, we extended the approach to protein-conditioned ligand generation. Immediate future work will focus on extending the model to support protein dynamics with minimal modifications, though this will require careful data preparation and extensive training.

# Acknowledgments

This work was partially supported by the Wallenberg AI, Autonomous Systems and Software Program (WASP) funded by the Knut and Alice Wallenberg Foundation and acknowledge the access to the WARA-Ops portal. We acknowledge the EuroHPC Joint Undertaking for awarding this project access to the EuroHPC supercomputer LUMI, hosted by CSC (Finland) and the LUMI consortium through a EuroHPC AI and Data-Intensive Applications call.

# Impact statement

This paper presents work whose goal is to advance the field of machine learning. There are many potential societal consequences of our work, none of which we feel must be specifically highlighted here.

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

# Appendix

# A. FlexiFlow

This section provides additional details about the architecture and training setup of FlexiFlow. Moreover, we provide additional information about metrics, datasets used to train the models, processing of the data and models hyperparameters. Because it is not possible to adopt the exact same notation, we instead use a notation closely aligned with that of SemlaFlow (Irwin et al., 2025), with the aim of helping the reader more clearly discern the key architectural differences.

## A.1. FlexiFlow Model in Details

We provide in the following sections a description of each module of the FlexiFlow architecture. The full architecture can be summarized as the composition of featurization, $L$ repeated FlexiFlow layers, and a feature refinement layer:

$$\text{FlexiFlow} = \underbrace{\text{Refinement}}_{L+2} \circ \underbrace{\text{FlexiFlowLayer} \circ \cdots \circ \text{FlexiFlowLayer}}_{L \text{ times}} \circ \underbrace{\text{Featurization}}_{1}. \qquad (15)$$

**Featurization layer.** Similarly to (Irwin et al., 2025), we process the input $h \in \mathbb{R}^{n \times |\mathcal{A}| \times |\mathcal{C}|}$, $e \in \mathbb{R}^{n \times n \times "\mathcal{B}|}$, $x \in \mathbb{R}^{n \times 3}$ and $y \in \mathbb{R}^{n \times 3}$ using a featurization layer that maps the input features into a higher dimensional space $\mathbb{R}^d$. $\mathcal{A}$, $\mathcal{B}$, and $\mathcal{C}$ represent the sets of atom, bond, and charge types, respectively. To this end, we use two different MLPs for the invariant features $h$ and $e$, while we use a shared linear layer for the coordinates $x$ and $y$. Specifically, the coordinates are first reshaped into $\mathbb{R}^{n \times 1 \times 3}$ and then projected into a higher dimensional space $\mathbb{R}^{n \times d \times 3}$. According to (Irwin et al., 2025) this operation helps to increase the expressivity of the model. For notation consistency $h$ and $e$ are cloned into $h^x$, $h^y$ and $e^x$, $e^y$ respectively, where $h^x$ and $e^x$ are used to update the features of $x$ and $h^y$ and $e^y$ for $y$.

**FlexiFlow layer.** Each FlexiFlow layer consists of a feed-forward block followed by a graph attention block:

$$\text{FlexiFlowLayer} = \mathcal{G}_{\text{attn}} \circ \mathcal{F}. \qquad (16)$$

**Feed-forward:** The features $h^x$, $h^y$, $e^x$, $e^x$, $x$ and $y$ are now now normalized as follows:

$$\tilde{x} = \phi_{\text{equi}}(x) \quad \tilde{y} = \phi_{\text{equi}}(y) \quad \tilde{h}^x = \phi_{\text{inv}}(h^x) \quad \tilde{h}^y = \phi_{\text{inv}}(h^y) \qquad (17)$$

where $\phi_{\text{equi}}(\cdot)$ and $\phi_{\text{inv}}(\cdot)$ are the equivariant and invariant normalization layers, from (Vignac et al., 2023) and (Ba et al., 2016), respectively. Subsequently, we use the normalized features to update the invariant features $h^x$ and $h^y$. These are updated as follows:

$$h_i^{x\,\text{ff}} = h_i^x + \Phi_\theta\big(\big[\tilde{h}_i^x, \|\tilde{x}_i\|\big]\big) \qquad h_i^{y\,\text{ff}} = h_i^y + \Phi_\theta\big(\big[\tilde{h}_i^y, \|\tilde{y}_i\|\big]\big), \qquad (18)$$

where $[\cdot, \cdot]$ denotes concatenation and the coordinate norm is taken component-wise:

$$\|\tilde{x}_i\| = [\|\tilde{x}_{i,1}\|, \ldots, \|\tilde{x}_{i,d}\|], \qquad (19)$$

and $\Phi_\theta$ is an MLP with this structure $\Phi_\theta(z) = W_2 \cdot \text{SiLU}(W_1 z + b_1) + b_2$ that maps back to the original dimensionality the features $\Phi_\theta : \mathbb{R}^{d+s} \to \mathbb{R}^d$. Since $h^{x\,\text{ff}}$ and $h^{y\,\text{ff}}$ are updated based on $\tilde{x}$ and $\tilde{y}$, respectively, the update of $h^{x,\text{ff}}$ is influenced by $x$ features, while the update of $h^{y,\text{ff}}$ is influenced by $y$ features.

To update the equivariant features $x$ and $y$,

$$x_i^{\text{ff}} = x_i + W_g\Big(\sum_{j=1}^d (W_f\,\tilde{x}_j) \otimes \Psi_\theta(\tilde{h}_i^x)\Big) \quad y_i^{\text{ff}} = y_i + W_g\Big(\sum_{j=1}^d (W_f\,\tilde{y}_j) \otimes \Psi_\theta(\tilde{h}_i^y)\Big), \qquad (20)$$

where $\otimes$ denotes the outer product, $\Psi_\theta : \mathbb{R}^d \to \mathbb{R}^p$ is and MLP defined as: $\Psi_\theta(z) = V_2 \cdot \text{SiLU}(V_1 \cdot z + c_1) + c_2$, $W_g$ and $W_f$ are two linear projections and $p$ is the projection dimension.

We summarize the feed-forward block with the compact for as $\mathcal{F}$.

**Graph attention:** The graph attention layer aims to combine invariant and equivariant features with the attention mechanism to update the nodes features representations. This module is important to let the $y$ features be influenced by $x$ features. In this paragraph we discuss the two components which is made of, the message computation and the attention mechanism, where the last is subdivided in two parts, the invariant attention and the equivariant attention.

*Message computation:* To compute the messages we first normalize the coordinates and invariant features using the same normalization scheme as in the feed forward layer:

$$\tilde{x} = \phi_{\text{equi}}(x) \quad \tilde{y} = \phi_{\text{equi}}(y) \quad \tilde{h}_x = \phi_{\text{inv}}(h^x) \quad \tilde{h}_y = \phi_{\text{inv}}(h^y) \tag{21}$$

Then, we perform the $\cdot$ product between the normalized coordinates to obtain $x_{\text{pairs}}$ and $y_{\text{pairs}}$. We also project the invariant features using a linear transformation obtaining $\hat{h}_x$ and $\hat{h}_y$. The messages are thus computed as follows:

$$x_p = \tilde{x}_i^{\text{ff}} \cdot \tilde{x}_j^{\text{ff } T}, \quad y_p = \tilde{y}_i^{\text{ff}} \cdot \tilde{y}_j^{\text{ff } T}, \quad h_p^x = [W_g \tilde{h}_i^{x\,\text{ff}} \| W_g \tilde{h}_j^{x\,\text{ff}}], \quad h_p^y = [W_g \tilde{h}_i^{y\,\text{ff}} \| W_g \tilde{h}_j^{y\,\text{ff}}] \tag{22}$$

Finally, we concatenate the features as reported below to obtain the final messages for $x$ and $y$:

$$\omega_p^x = [h_p^x, x_p, e^x], \quad \omega_p^y = [h_p^x \cdot h_p^y, x_p, y_p, e^x \cdot e^y] \tag{23}$$

where $W_g$ represents a linear projection. The final messages are computed using two separate MLPs for $\text{MLP}_x$ and $\text{MLP}_y$ from $\omega_p^x$ and $\omega_p^y$ as they have different input shapes, obtaining the messages that are used to update the $x$, $y$, $h^x$, $h^y$, $e^x$ and $e^y$ features in the equivariant and invariant attention blocks. These are now denoted as $\omega_{p,h}^x$, $\omega_{p,h}^y$, $\omega_{p,x}^x$, $\omega_{p,y}^y$, $\omega_{p,e}^x$ and $\omega_{p,e}^y$.

*Invariant Attention:* We proceed updating the $h^x$ and $h^y$ features separately as outlined below, where $\sigma$ denote the following operation $\sigma(\mathbf{z})_i = e^{z_i} / \sum_{j=1}^K e^{z_j}$ applied on the penultimate feature dimension:

$$\alpha_x^k = \sigma(\omega_{p,h}^x) \quad \alpha_y^k = \sigma(\omega_{p,h}^y) \tag{24}$$

$$\tilde{h}^x = W_v \phi_{\text{inv}}(h^x) \quad \tilde{h}^y = W_v \phi_{\text{inv}}(h^y) \quad \tilde{h}^k \in \mathbb{R}^{N \times d\text{head}} \text{ for } k = 1, \dots, n_{\text{heads}} \tag{25}$$

Thus, we apply the same process to aggregate the features for each head as reported below:

$$h_{\text{aggr}}^k = \sum_j \alpha_{ij}^k \cdot \tilde{h}_j^k \quad w_i^k = \sqrt{\sum_j (\alpha_{ij}^k)^2} \quad h^{\text{message}} = W_z [h_{\text{aggr}}^k \cdot w_i^k] \tag{26}$$

where $W_v : \mathbb{R}^d \to \mathbb{R}^m$ and $W_z : \mathbb{R}^m \to \mathbb{R}^d$ are linear projections and $m$ is the latent message dimensionality.

*Equivariant Attention:* We now update the coordinate features $x^{\text{ff}}$ and $y^{\text{ff}}$ using and equivariant attention mechanism following (Irwin et al., 2025). Similarly to the invariant attention, we compute the attention weights as follows:

$$\alpha_x^k = \sigma(\omega_{p,x}^x) \quad \alpha_y^k = \sigma(\omega_{p,y}^y) \tag{27}$$

$$\tilde{\alpha}^k = W_r \alpha^k \quad \tilde{x}_k = W_r \phi_{\text{equi}}(x_k) \quad \tilde{y}_k = W_{e^1} \phi_{\text{equi}}(y_k) \tag{28}$$

The same attention strategy is applied to $\tilde{x}$ and $\tilde{y}$ to aggregate the contribution of each node to the final message update:

$$x_{\text{aggr}}^k = \sum_j^N \tilde{\alpha}_{ij}^k \cdot \left( \frac{\tilde{x}_i - \tilde{x}_j}{(\tilde{x}_i - \tilde{x}_j + \epsilon)} \right) \quad w_i^k = \sqrt{\sum_j (\tilde{\alpha}_{ij}^k)^2} \quad x^{\text{message}} = W_s [x_{\text{aggr}}^k \cdot w_i^k] \tag{29}$$

where $\epsilon$ is equal to $10^{-12}$ and $W_r : \mathbb{R}^d \to \mathbb{R}^m$ and $W_s : \mathbb{R}^m \to \mathbb{R}^d$ are two linear projections and $m$ is the latent message dimensionality. Thus, we finally update the features as follows using this scheme:

$$x = x + x^{\text{message}} \quad y = y + y^{\text{message}} \tag{30}$$

$$h^x = h^x + h^{x,\text{message}} \quad h^y = h^y + h^{y,\text{message}} \tag{31}$$

$$e^x = e^x + \omega_{p,e}^x \quad e^y = e^y + \omega_{p,e}^y \tag{32}$$

We now refer to one graph attention block forward pass with this notation $\mathcal{G}_{attn}$.

**Features refinement.** We apply a final feed forward layer $x, y, h^x, h^y$ and use the output to shift the representations at the last FlexiFlow layer. On $x$ and $y$ we use the same equivariant norm, and a linear transformation to shrink the coordinate sets into one. On the edge features we apply the Edge Update Layer before the final MLP projection, see the dedicated Appendix section A.2. Subsequently we apply two different invariant normalization layers to $h^x$ (atoms, charges) and $e^x$ (bonds), and use three separate MLP to obtain the logits for the atoms, charges and bonds types. These are finally projected into $\mathbb{R}^{n \times |\mathcal{A}|}$, $\mathbb{R}^{n \times |\mathcal{C}|}$ and $\mathbb{R}^{n \times n \times |\mathcal{B}|}$, where $|\mathcal{A}|$, $|\mathcal{C}|$ and $|\mathcal{B}|$ are the cardinality of the sets $\mathcal{A}, \mathcal{C}$ and $\mathcal{B}$ for atom, charge and bond types.

**Protein conditioning.** We model a protein as a set of invariant and equivariant features for each protein atom, from one-hot encoded $\rho^{inv} \in \mathbb{R}^{\ell \times |\mathcal{H}|}$ and coordinates $\rho^{equi} \in \mathbb{R}^{\ell \times 3}$ where $\ell$ is the number of protein atoms and $\mathcal{H}$ is the set of protein atom types. To support the conditioning on the protein features, we added a protein feed forward layer before the graph attention layer in each FlexiFlow layer. Each protein atom invariant and equivariant features set of $\rho^{inv}$ and $\rho^{equi}$ are than fed into the graph attention layer, where we separately compute the messages between ligand-ligand and ligand-protein and concatenate them before the attention mechanism. Further details are provided below are provided in Appendix A.3.

### A.2. Edge Feature Refinement Layer FlexiFlow

Similarly to (Irwin et al., 2025), we postprocess the features of the bonds (edges) using a dedicated layer, which we call Edge Update Layer (EUL). The EUL takes as input the 3D coordinat sets of the atoms (nodes) $x \in \mathbb{R}^{n \times d \times 3}$, their features $h \in \mathbb{R}^{n \times d}$ and the bond features $e \in \mathbb{R}^{n \times n \times d}$. It outputs the updated bond features $e'$.

We normalize $x$, $h$ and $e$ using the equivariant and invariant normalizations $\phi_{\text{equi}}$ and $\phi_{\text{inv}}$ respectively:

$$\tilde{x} = \phi_{\text{equi}}(x) \quad \tilde{h} = \phi_{\text{inv}}(h^x) \quad \tilde{e} = \phi_{\text{inv}}(e^x) \tag{33}$$

Over the coordinate sets, we compute the geometric distances and inner products, and then concatenate them:

$$\Delta_{ij} = \tilde{x}_i - \tilde{x}_j, \qquad d_{bij} = \|\Delta_{ij}\|_2^2, \qquad p_{ij} = \langle \tilde{x}_i, \tilde{x}_j \rangle, \tag{34}$$

$$\tag{35}$$

Subsequently, we apply a linear layer to project the node features to the message dimension $d$:

$$\tilde{h}_i = W_h \tilde{h}_i + b_h, \quad W_h \in \mathbb{R}^{d \times d}. \tag{36}$$

and we form the pair features by concatenation:

$$h_{ij}^{\text{pair}} = [\tilde{h}_i \,\|\, \tilde{h}_j] \in \mathbb{R}^{2d}. \tag{37}$$

Finally, we concatenate all the features to form the input to the message MLP:

$$F_{ij} = [\, h_{ij}^{\text{pair}} \,\|\, d_{ij} \,\|\, p_{ij} \,\|\, \tilde{e}_{ij} \,] \in \mathbb{R}^{2d+2+d} \tag{38}$$

$$m_{ij} = \sigma(W_l F_{ij} + b_1), \tag{39}$$

$$e'_{ij} = W_q m_{ij} + b_2, \tag{40}$$

where $\sigma$ is the SiLU activation function, $W_l \in \mathbb{R}^{d \times (2d+2+d)}$, $W_q \in \mathbb{R}^{d \times d}$ and $e' \in \mathbb{R}^{n \times n \times d}$ are the updated bond features.

### A.3. Protein Conditioning

This section provides additional about how the ligand-protein interaction conditioning is computed. After Equation 13, we use the normalized features coordinates $\tilde{x}$ of the ligand, and normalize the protein coordinate features $\rho_{ff}^{equi}$, protein atoms features $\rho_{ff}^{inv}$ and ligand atom features $h^x$ as follows:

$$\tilde{\rho}^{equi} = \phi_{\text{equi}}\big(\rho_{ff}^{equi}\big), \qquad \tilde{\rho}^{inv} = \phi_{\text{inv}}\big(\rho_{ff}^{inv}\big), \qquad \tilde{h} = \phi_{\text{inv}}(h^x). \tag{41}$$

We concatenate the protein and ligand atoms features pairwise:

$$h_{ij}^{\rho^{inv}} = [\, \tilde{h}_i \,\|\, \tilde{\rho}_j^{inv} \,] \in \mathbb{R}^{2d}. \tag{42}$$

Next, we compute the distances between the ligand and protein atoms:

$$d_{ijs} = \sqrt{\|\tilde{x}_{is} - \tilde{\rho}_{js}^{equi} + \varepsilon\|_2^2}. \tag{43}$$

where $\varepsilon$ is equal to $1e - 12$. Lastly, we concatenate the distances to the pair features and apply an MLP to project them to the desired dimension of the message:

$$m_{ij}^{\rho} = [\, h_{ij}^{\rho^{inv}} \ \| \ D_{ij} \,] \in \mathbb{R}^{(2d+s)}, \tag{44}$$

$$m_{ij}^{\rho} = W_o \, \sigma\big(W_a m_{ij}^{\rho} + b_a\big) + b_o \quad \in \mathbb{R}^d, \tag{45}$$

with $W_a \in \mathbb{R}^{d \times (2d+s)}$, $W_o \in \mathbb{R}^{d \times d}$, $\sigma$ the SiLU activation, while $b_a$ and $b_o$ are the respective bias terms.

The message is computed separately for $x$ and $y$ coordinate set, using the same weights and protein features. Finally, we concatenate the $x$ and $y$ coordinate, atoms and message features with the protein coordinate, atoms and message features, respectively, and follow the same steps to perform the features update according with Equation 26 and 29.

## A.4. Loss Setting FlexiFlow

To support the generation of the graphs $x$ and $\mathcal{S}$, our loss follow this scheme:

$$\mathcal{L} = \mathcal{L}_{x,y} + \mathcal{L}_a + \mathcal{L}_c + \mathcal{L}_e + \mathcal{L}_{\text{reg}} \tag{46}$$

where (1) $\mathcal{L}_{x,y}$ coordinates loss for $x$ and $y$ is defined as the mean squared error between the predicted and target coordinates:

$$\mathcal{L}_{x,y} = \frac{1}{N} \sum_{i=1}^{N} \|\hat{x}_i^{(1)} - x_i^{(1)}\|^2 + \frac{1}{N} \sum_{i=1}^{N} \|\hat{y}_i^{(1)} - y_i^{(1)}\|^2 \tag{47}$$

where $\hat{x}_i^{(1)}$ and $\hat{y}_i^{(1)}$ are the predicted coordinates for the $i$-th atom in $x$ and $y$ respectively, while $x_i^{(1)}$ and $y_i^{(1)}$ are the ground truth coordinates and $N$ the number of atoms. (2) $\mathcal{L}_a, \mathcal{L}_c, \mathcal{L}_e$ are the negative log-likelihood losses for the categorical atoms, charges and bonds types respectively:

$$\mathcal{L}_a = -\frac{1}{n} \sum_{i=1}^{n} \log p(\hat{a}_i^{(1)} = a_i^{(1)}), \tag{48}$$

$$\mathcal{L}_c = -\frac{1}{n} \sum_{i=1}^{n} \log p(\hat{c}_i^{(1)} = c_i^{(1)}), \tag{49}$$

$$\mathcal{L}_e = -\frac{1}{n^2} \sum_{(i,j) \in \mathcal{E}} \log p(\hat{e}_{ij}^{(1)} = e_{ij}^{(1)}) \tag{50}$$

where $\hat{a^{(1)}}_i, \hat{c}_i^{(1)}$ and $\hat{e}_{ij}^{(1)}$ are the predicted atom type, charge and bond type for the $i$-th atom and $(i,j)$-th bond respectively, while $a_i^{(1)}, c_i^{(1)}$ and $e_{ij}^{(1)}$ are the ground truth values and $\mathcal{E}$ is the set of edges in the molecular graph. (3) $\mathcal{L}_{\text{reg}}$ a regularization loss is made by different components: (3.1) enforce bonds lengths consistency on both $x$ and $y$:

$$D_{ij}^{x,1} = \|x_i^{(1)} - x_j^{(1)}\|_2, \qquad D_{ij}^{x,p} = \|\hat{x}_i - \hat{x}_j\|_2, \tag{51}$$

$$D_{ij}^{y,1} = \|y_i^{(1)} - y_j^{(1)}\|_2 \, P_{ij}, \qquad D_{ij}^{y,p} = \|\hat{y}_i - \hat{y}_j\|_2 \, P_{ij}. \tag{52}$$

where $x_i^{(1)}, y_i^{(1)}$ be target (ground-truth) coordinates and $\hat{x}_i, \hat{y}_i$ the predicted coordinates and $D$ the respective pairwise distances.

Let $e_{ijk}^{(1)}$ be the (target) bond-type logits and define the bond presence mask

$$B_{bij} = \mathbf{1}\left[\arg\max_k e_{ijk}^{(1)} > 0\right], \tag{53}$$

so only pairs with a non-zero bond type contribute. The adjacency (distance) constraint losses implemented in the code are the mean absolute deviations over all pairs:

$$\mathcal{L}^x_{\text{adj}} = \frac{1}{N^2} \sum_{i,j=1}^{N} B_{ij} \left| D^{x,p}_{ij} - D^{x,1}_{ij} \right|, \qquad \mathcal{L}^y_{\text{adj}} = \frac{1}{N^2} \sum_{i,j=1}^{N} B_{ij} \left| D^{y,p}_{ij} - D^{y,1}_{ij} \right|. \tag{54}$$

Additionally, (3.2) we align the categorical for $y$ on $x$:

$$\mathcal{L}_{\text{bond-align}} = \left( \frac{\sum_{i}^{n} \sum_{j}^{n} \| \hat{E}^x_{ij} - \hat{E}^y_{ij} \|^2_2}{\sqrt{n^2} + \varepsilon} \right) - \frac{1}{n^2} \sum_{(i,j) \in \mathcal{E}} \log p(\hat{e}^y_{ij} = e_{ij}), \tag{55}$$

$$\mathcal{L}_{\text{type-align}} = \left( \frac{\sum_{i} \| \hat{H}^x_i - \hat{H}^y_i \|^2_2}{\sqrt{n} + \varepsilon} \right) - \frac{1}{n} \sum_{i=1}^{n} \log p(\hat{a}^y_i = a_i), \tag{56}$$

$$\mathcal{L}_{\text{charge-align}} = \left( \frac{\sum_{i} \| \hat{C}^x_i - \hat{C}^y_i \|^2_2}{\sqrt{n} + \varepsilon} \right) - \frac{1}{n} \sum_{i=1}^{n} \log p(\hat{c}^y_i = c_i). \tag{57}$$

where $\varepsilon$ is a small constant to avoid division by zero, $\hat{E}^x_{ij}, \hat{E}^y_{ij}$ are the predicted bond type logits for the $(i,j)$-th bond in $x$ and $y$ respectively, while $\hat{H}^x_i, \hat{H}^y_i$ and $\hat{C}^x_i, \hat{C}^y_i$ are the predicted atom type and charge logits for the $i$-th atom in $x$ and $y$ respectively. The full regularization loss is thus defined as:

$$\mathcal{L}_{\text{reg}} = \mathcal{L}^x_{\text{adj}} + \mathcal{L}^y_{\text{adj}} + \mathcal{L}_{\text{bond-align}} + \mathcal{L}_{\text{type-align}} + \mathcal{L}_{\text{charge-align}}. \tag{58}$$

### A.5. Training Scheme & Interpolants Setting

Let $\mathcal{A}$, $\mathcal{B}$, and $\mathcal{C}$ denote the sets of atom types, bond types, and charge types, respectively. We denote by $(a, b, c) \sim \text{Cat}(1/|\mathcal{A}|) \cdot \text{Cat}(1/|\mathcal{B}|) \cdot \text{Cat}(1/|\mathcal{C}|)$ the sampling of a triplet from three independent categorical distributions, each uniform over its corresponding set. We draw samples from our time-dependent categorical distribution following the approach of (Campbell et al., 2024). Specifically, the time variable $t$ is sampled from a Beta distribution with parameters $\alpha = 2.0$ and $\beta = 1.0$. During training, given samples from the noised data distribution, the objective is to predict the corresponding target data distribution. The coordinate interpolants $x_t$ and $y_t$ are obtained following (Tong et al., 2024).

---

**Algorithm 2** Training scheme

---

1: $(x_1, y_1, a_1, b_1, c_1) \sim p_{data}$
2: $x_0 \sim \mathcal{N}(0, \mathrm{I})$, $y_0 \sim \mathcal{N}(0, \mathrm{I})$, $t \sim \text{Beta}(\alpha, \beta)$
3: $x_t \sim \mathcal{N}(tx_1 + (1-t)x_0, \sigma^2)$
4: $y_t \sim \mathcal{N}(ty_1 + (1-t)y_0, \sigma^2)$
5: $a_0, b_0, c_0 \sim \text{Cat}(1/|\mathcal{A}|) \cdot \text{Cat}(1/|\mathcal{B}|) \cdot \text{Cat}(1/|\mathcal{C}|)$,
6: $a_t, b_t, c_t \sim \text{CatInterp}(t, a_0, a_1) \cdot \text{CatInterp}(t, b_0, b_1) \cdot \text{CatInterp}(t, c_0, c_1)$,
7: **while** Training **do:**
8: $\quad (\hat{x}_1, \hat{y}_1, \hat{a}^x_1, \hat{b}^x_1, \hat{c}^x_1, \hat{a}^y_1, \hat{b}^y_1, \hat{c}^y_1) \leftarrow f_\theta(x_t, y_t, a_t, b_t, c_t)$
9: $\quad \mathcal{L}(\theta) = \mathcal{L}_{x,y}(\hat{x}_1, x_1, \hat{y}_1, y_1) + \mathcal{L}_a(\hat{a}^x_1, \hat{a}^y_1, a_1) + L_c(\hat{c}^x_1, \hat{c}^y_1, c_1) +$
10: $\quad\quad\quad \mathcal{L}_b(\hat{b}^x_1, \hat{b}^y_1, b_1) + L_{reg}(\hat{x}_1, x_1, \hat{y}_1, y_1)$
11: **end while**

---

### A.6. Model & Training Hyperparameters

Here we provide the list of hyperparameters that are kept fixed across the models configurations: dimension edge features = 128 and invariant positional embedding size = 64. In Table 3 are reported the parameters that vary in the model configuration.

Refer to (Irwin et al., 2025) for further details. All the result in the paper that do not specifically mention the model size use the Large configuration of it.

| Model type | n_layers | d_model | d_message & d_message_hidden | n_attn_heads | Parameters |
|---|---|---|---|---|---|
| Small (S) | 6 | 384 | 64 | 12 | 17.2M |
| Medium (M) | 8 | 384 | 128 | 32 | 24.7M |
| Large (L) | 12 | 384 | 128 | 32 | 37.7M |

*Table 3.* The table reports the parameters that vary across model configurations, while all other fixed parameters are listed below. d stands for model features dimensionality.

Key training configuration details:

- training seed = 42
- coordinate noise $\sigma = 0.2$ on the interpolated coordinates
- Adam (Kingma & Ba, 2015) with learning rate lr=$1e-3$ and weight decay $0.0$
- LinearLR is used as learning rate scheduler with start_factor=$1e-2$ and total_iters=10000
- all the models are trained using exponential moving average (EMA)

### A.7. Data Pre-processing Molecular Structures

**QM9.** The QM9 dataset (Ramakrishnan et al., 2014) consists of ∼134k small organic molecules with up to 9 heavy atoms (C, O, N, F) and their corresponding 3D conformations. We follow the standard split used in previous works (Hoogeboom et al., 2022; Vignac et al., 2023), using $\sim 100k$ molecules for training. We preprocess the data following the steps outlined in (Irwin et al., 2025), which include centering the molecules at the origin, normalizing the coordinates, checking validity of the molecular graphs and graph fragmentation with RdKit. Since the hydrogen atoms are kept, the resulting model vocabulary is composed by (H, C, N, O, F).

Since QM9 provides only one conformer per molecule, we augment the dataset by generating 20 conformers per molecule using the ETKDG method (Riniker & Landrum, 2015) implemented in RdKit (obtaining ∼1.8M total samples). We then optimize these conformers using the MMFF94 force field (Halgren, 1996) to ensure physically plausible structures. The target $x$ conformer is selected as the closest conformation to the mean, while the remaining conformers form the set $\mathcal{S}$.

**GEOM Drugs.** The GEOM Drugs dataset (Axelrod & Gómez-Bombarelli, 2022) contains ∼400k unique drug-like molecules with up to 181 atoms (H, B, C, N, O, F, Si, P, S, Cl, Br, I) and their corresponding 3D conformations. Conversely to QM9, GEOM Drugs provides multiple conformers per molecule, with an average of 21 conformers per molecule. To achieve these conformers, the authors used GFN2-xTB calculations (Bannwarth et al., 2019), a physics-based method that provides accurate low-energy conformers.

Similarly to QM9, we selected the target $x$ conformer as the closest conformation to the mean, while the remaining conformers form the set $\mathcal{S}$. We use the same splits as in previous works (Vignac et al., 2023; Le et al., 2024), with $\sim 5.8M$ molecules for training. The data are processed similarly to QM9, ensuring that the molecules are centered, normalized, and valid. The model vocabulary comprises (H, B, C, N, O, F, Si, P, S, Cl, Br, I), with hydrogens retained during training. Our final training set contains 243,718 unique molecular target graphs $x$ and 5,491,198 distinct molecular conformers $y$.

### A.8. Metrics for De-novo Generation

We provide a description of the metrics used for 3D generation, specifically, validity, uniqueness, novelty, atom stability, and molecule stability.

* **Validity:** Validity measures the percentage of generated molecules that are chemically valid according with the sanitization check done by RDKit. A molecule is considered valid if it passes the sanitization check, which includes checks for valence, aromaticity, and other.

* **Uniqueness:** Uniqueness measures the percentage of unique molecular graphs converted into canonical SMILES strings.

* **Novelty:** Novelty measures the percentage of generated molecules that are not present in the training set.

* **Atom Stability:** Atom stability measures the percentage of atoms in the generated molecules that have a valid valence according to their element type.

* **Molecule Stability:** Molecule stability measures the percentage of generated molecules where all atoms are stable.

* **Energy:** According to the Boltzmann distribution, the probability of a conformation $x$ is determined by its energy $U(x)$ as $P(x_i) = Z^{-1}(e^{-U(x_i)/k_B T})$, where $U(x_i)$ is the energy of state $i$ parametrized by the MMFF96 force field, $k_B$ is the Boltzmann constant, $T$ is the absolute temperature, $Z$ is the partition function.

* **Strain:** The strain is computed as $U(x) - U(x^*)$, where $x$ is a conformation and $x^*$ is the energy minimized conformer.

### A.9. Metrics for Conformer Generation

Similarly to (Ganea et al., 2021), (Jing et al., 2022) and (Havens et al., 2025), we compute Average Minimum RMSD (AMR) and Coverage (Cov) for Precision (P) and Recall (R).

We generate with FlexiFlow a number $N$ of conformers for each molecule generated from scratch, then we use the conformer with the lowest energy as the input to CREST, which generates a set of $M$ reference conformers, where $M < N$. Equivalently, to generate the conformers with RDKit, we use the conformer with the lowest energy generated by FlexiFlow as the input to RDKit to produce a set of $N$ conformers.

Lastly, we compare against the reference CREST conformers to the generated ones. Finally, we use the RMSD metric to compare generated conformers to reference conformers, which aims to capture both the quality and diversity of the generated conformers. The RMSD is computed as the minimum distance between two conformers, taking into account the molecular structure.

R stands for recall, P for precision, Cov for coverage, and AMR for average minimum RMSD. In this context, recall measures the coverage of the generated conformers against the reference conformers, while precision measures how closely at least one of the generated conformers approximates a reference conformer.

$$\text{Cov-R}(\delta) := \frac{1}{M} \left| \{ m \in \{1, \ldots, M\} : \exists n \in \{1, \ldots, N\}, \quad \text{RMSD}(C_n, C_m) < \delta \} \right| \tag{59}$$

$$\text{AMR-R} := \frac{1}{M} \sum_{m \in \{1, \ldots, M\}} \min_{n \in \{1, \ldots, N\}} \text{RMSD}(C_n, C_m) \tag{60}$$

$$\text{Cov-P}(\delta) := \frac{1}{N} \left| \{ n \in \{1, \ldots, N\} : \exists m \in \{1, \ldots, M\}, \quad \text{RMSD}(C_n, C_m) < \delta \} \right| \tag{61}$$

$$\text{AMR-P} := \frac{1}{N} \sum_{n \in \{1, \ldots, N\}} \min_{m \in \{1, \ldots, M\}} \text{RMSD}(C_n, C_m) \tag{62}$$

where $\delta > 0$ is the coverage threshold.

### A.10. RMSD After Energy Minimization Metric

We perform additional minimization until convergence (within a fixed number of minimization steps) using a physico-chemical force field (in this case MMFF94). Thus, an RMSD $> 0$ indicates that we have two conformers $\mathbf{x}_1$ and $\mathbf{x}_2$ such that $\mathbf{x}_1 \neq \mathbf{x}_2$ and

$$\nabla E(\mathbf{x}_1) \equiv \nabla E(\mathbf{x}_2) \equiv 0. \tag{63}$$

For both $\mathbf{x}_1$ and $\mathbf{x}_2$, there exist neighborhoods with radii $d_1$ and $d_2$ such that

$$f(\mathbf{x}_1) < f(\mathbf{x}) \quad \text{for all } \mathbf{x} \text{ satisfying } \|\mathbf{x} - \mathbf{x}_1\| < d_1, \tag{64}$$

and similarly for $\mathbf{x}_2$. This means they correspond to distinct local minima.

## B. Determinant Decomposition

As part of this section of the Appendix, we provide the proof of the determinant decomposition of the Jacobian of the flow $\psi_t(x, \mathcal{S})$ with respect to $(x, \mathcal{S})$ under local dependence. Moreover, we formulate the target velocity field $u_t(x, \mathcal{S} \mid x_1, \mathcal{S}_1)$ in closed form under linear interpolation on sets.

### B.1. Proof Determinant Decomposition

Let be $\mathcal{S} = \{y_1, \ldots, y_m\}$ and define $\psi_t(x, \mathcal{S})$ as the concatenation of the flows among $\mathcal{S}$

$$\psi_t(x, \mathcal{S}) := \left\| \begin{array}{c} \\ y \in \mathcal{S} \end{array} \right. \psi_t(x, y) = \left( \psi_t^{(1)}(x, y_1), \ldots, \psi_t^{(m)}(x, y_m) \right), \tag{65}$$

where each block $\psi_t^{(i)}(x, y_i) \in \mathbb{R}^{n \times m}$.

**Lemma 1:** We consider the Jacobian of the flow $\psi_t$ as:

$$J(x, \mathcal{S}) := \nabla_{(x, \mathcal{S})} \psi_t(x, \mathcal{S}). \tag{66}$$

where $J$ satisfies the local dependence property, $\frac{\partial \psi_t^i}{\partial y_j} = 0 \quad$ for $j \neq i$, such that each block depends only on $(x, y_i)$.

Under the local dependence assumption, we now reconduct the determinant of $J$ to a block diagonal matrix to allow the flow decomposition. For simplicity, we provide a proof for $m = 2$ and $n = 2$, however, the proof can be easily extended to $m > 2$. We start with three 2-dimensional vectors:

$$x = \begin{bmatrix} x_1 & x_2 \end{bmatrix}, \quad y^{(1)} = \begin{bmatrix} y_{1,1} & y_{1,2} \end{bmatrix}, \quad y^{(2)} = \begin{bmatrix} y_{2,1} & y_{2,2} \end{bmatrix}.$$

We concatenate the vectors as follows:

$$v_1 = \begin{bmatrix} x_1 & x_2 & y_{1,1} & y_{1,2} \end{bmatrix}, \qquad v_2 = \begin{bmatrix} x_1 & x_2 & y_{2,1} & y_{2,2} \end{bmatrix}$$
$$z = \begin{bmatrix} v_1 & v_2 \end{bmatrix} = \begin{bmatrix} x_1 & x_2 & y_{1,1} & y_{1,2} & x_1 & x_2 & y_{2,1} & y_{2,2} \end{bmatrix}.$$

Thus $z \in \mathbb{R}^8$ in this special case. Hence we can define the full flow as the concatenation of two independent flows:

$$\psi_t(z) = \begin{bmatrix} \psi_t^{(1)}(z) \\ \psi_t^{(2)}(z) \end{bmatrix}$$

where each $\psi_t^{(i)}(z) \in \mathbb{R}^4$ for $i = 1, 2$. Therefore, we can compute the Jacobian of the full flow $\psi_t(z)$ with respect to $z$ as:

$$J(z) = \frac{\partial \psi_t(z)}{\partial z} \in \mathbb{R}^{8 \times 8}.$$

The Jacobian has a block-diagonal structure:

$$J(z) = \begin{bmatrix} J^{(1)} & 0 \\ 0 & J^{(2)} \end{bmatrix},$$

where each block $J^{(i)} = \frac{\partial \psi_t^{(i)}(z)}{\partial z} \in \mathbb{R}^{4 \times 8}$ for $i = 1, 2$. This, for $J^{(1)}$, where $i = 1, \ldots, 4$,

$$\psi_t^{(1)}(z) = z_i^2 + \prod_{k=1}^{4} z_k. \tag{67}$$

Thus

$$\frac{\partial \psi_t^{(1)}}{\partial z_j} = \begin{cases} 2z_i + \prod_{\substack{k=1 \\ k \neq i}}^{4} z_k, & j = i, \\[2em] \prod_{\substack{k=1 \\ k \neq j}}^{4} z_k, & j \neq i, \; j \in \{1, 2, 3, 4\}, \\[2em] 0, & j \in \{5, 6, 7, 8\}. \end{cases} \tag{68}$$

For $J^{(2)}$, where $i = 5, \ldots, 8$,

$$\psi_t^{(2)}(z) = z_i^2 + \prod_{k=5}^{8} z_k. \tag{69}$$

Thus

$$\frac{\partial \psi_t^{(2)}}{\partial z_j} = \begin{cases} 2z_i + \prod\limits_{\substack{k=5 \\ k \neq i}}^{8} z_k, & j = i, \\ \prod\limits_{\substack{k=5 \\ k \neq j}}^{8} z_k, & j \neq i, \ j \in \{5, 6, 7, 8\}, \\ 0, & j \in \{1, 2, 3, 4\}. \end{cases} \tag{70}$$

As the Jacobian $J$ results in block diagonal matrix, it can be factorized as the product of block determinants. Therefore,

$$\det\big(\nabla_{(x,\mathcal{S})}\psi_t(x,\mathcal{S})\big) = \prod_{i=1}^{m} \det\big(\nabla_{(x,y_i)}\psi_t^i(x,y_i)\big). \tag{71}$$

### B.2. Target Vector Field $u_t$ for the Special Case of Linear Interpolants on Sets

We define the target velocity field as the conditional expectation of the trajectory velocity:

$$u_t(z \mid z_1) = \mathbb{E}[\dot{z}_t \mid z_t = z, \ z_1]. \tag{72}$$

where $\dot{z}_t$ correspond to the $z_t$ derivative. In our case, the path definition reduces to linear interpolation which can be defined as:

$$z_t = (1 - t)z_0 + tz_1, \tag{73}$$

Thus,

$$u_t(z \mid z_1) = \mathbb{E}[z_1 - z_0 \mid z_t = z, \ z_1], \tag{74}$$

and the path is a straight line that connects $z_0$ to $z_1$, hence the velocity is constant along the path. The extension to $(x, \mathcal{S})$ can be seen by reformulating $z = (x, \mathcal{S})$ and defining the path elementwise:

$$(x_t, y_{t,i}) = (1 - t)(x_0, y_{0,i}) + t(x_1, y_{1,i}), \quad i = 1, \ldots, m. \tag{75}$$

where $(x_0, \mathcal{S}_{0,i})$ is sampled from the prior distribution and $(x_1, \mathcal{S}_{1,i})$ from the data distribution and $i$ indicates the $i$-th element in the set $\mathcal{S}$. So $(x_t, \mathcal{S}_t) = \{(1 - t)(x_0, y_{0,i}) + t(x_1, y_{1,i})\}_{i=1}^{m}$. Now we can condition on $(x_t, \mathcal{S}_t) = (x, \mathcal{S})$ with endpoint $(x_1, \mathcal{S}_1)$:

$$u_t(x, \mathcal{S} \mid x_1, \mathcal{S}_1) = \mathbb{E}\Big[(x_1 - x_0, \{y_{1,i} - y_{0,i}\}_{i=1}^{m}) \,\Big|\, (x_t, \mathcal{S}_t) = (x, \mathcal{S}), (x_1, \mathcal{S}_1)\Big]. \tag{76}$$

# C. Additional Results

## C.1. Additional Results Top-k Fraction Generated Conformers QM9 and GEOM Drugs

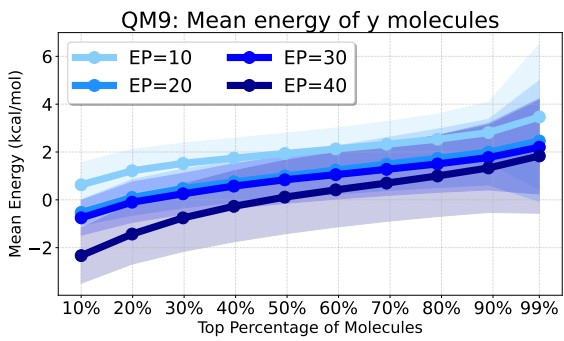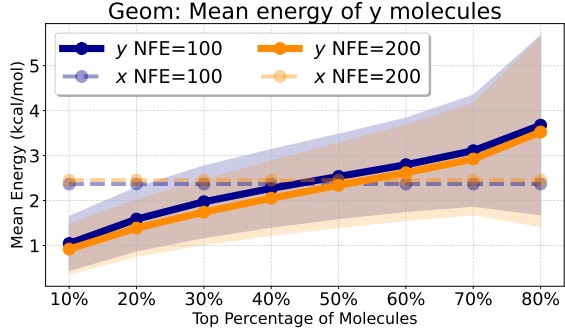

*Figure 7.* Figures show the energy per atom computed with MMFF94 force field on the top-$k$ fraction of $y$ conformers per $x$ reference molecule generated. The trend is shown on QM9 varying the epochs (left) and GEOM Drugs varying number of inference steps (NFE) reporting trends on $x$ and $y$.

**Normalized energies of FlexiFlow generated conformers.** In this setting, we use MMFF94 (Halgren, 1996) as the energy function for conformers, normalized by the number of atoms (kcal/mol/atom). We sampled 100 molecules with 300 conformations per molecule from both GEOM Drugs and QM9. Figures 7 show the mean conformer energy for each molecule: QM9 on the left and GEOM Drugs on the right. The x-axis in both plots reports the percentage of top-$k$ molecules ranked by energy. For QM9, we observe that low-energy conformers are concentrated within the top 30%, with only marginal improvements beyond this range. Energies also decrease as the number of training epochs increases. For GEOM Drugs, the energies of both the representative conformer and the remaining conformers remain stable across inference steps. Since the training reference conformer $x$ is chosen as the one closest to the average conformation within the set, its energy is expected to be near the mean. This is confirmed in the figure: the energy of $x$ is around 2.2 kcal/mol/atom, while the energies along $y$ range from 1 to 3 kcal/mol/atom.

**C.2. Additional Energies QM9**

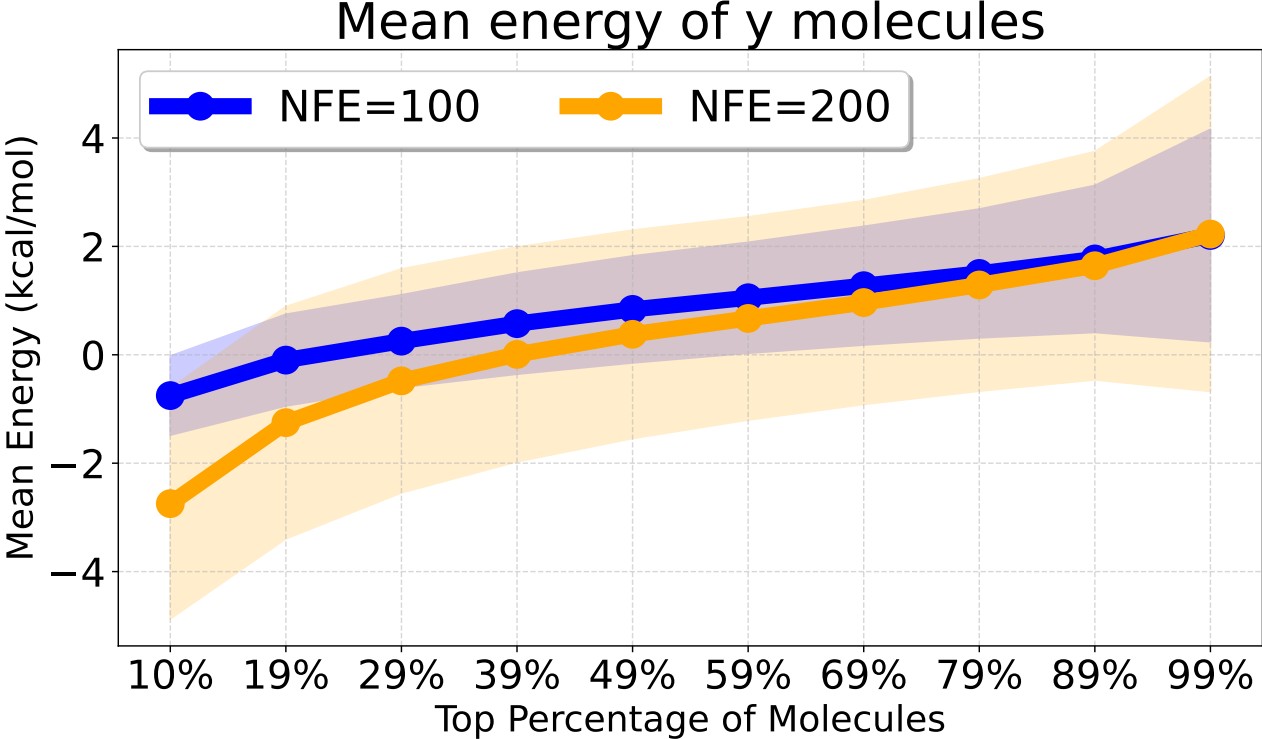

*Figure 8.* The conformers are sorted based on their energy values, and we plot mean and standard deviation for the top-$k$ fraction of conformers generated. The results are reported by sampling 300 conformers per molecule on 100 molecules using the base model trained on QM9 for 40 epochs. In figure we can observe that using more denoising steps (NFE), from 100 to 200 steps, during inference leads to lower energy values. However, the cost of sampling with more steps is linearly higher.

**C.3. Additional Results Conformers QM9**

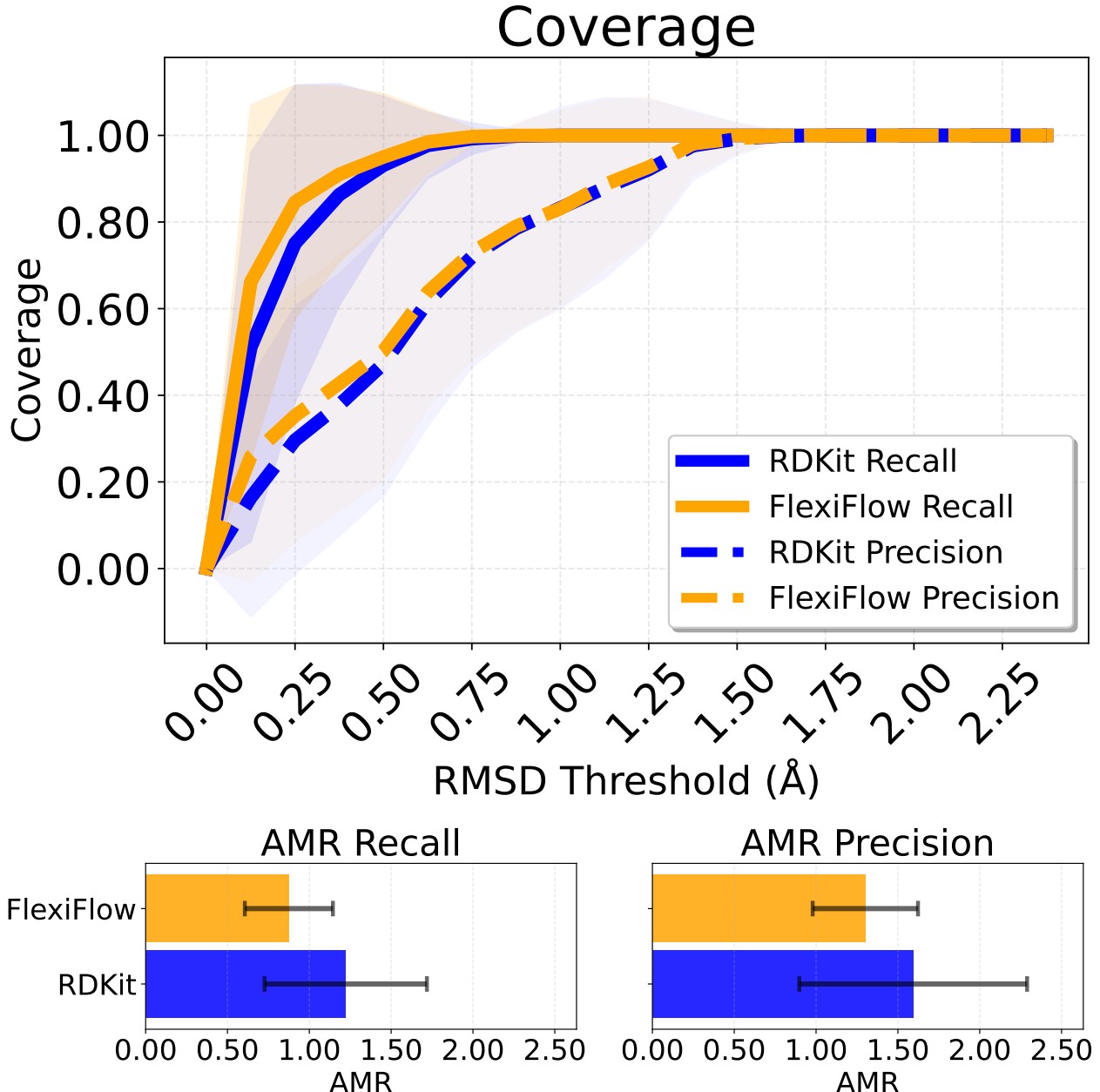

*Figure 9.* Coverage (top) and AMR (down) precision and recall for conformer generation on QM9. As the target data distribution of the conformers is derived from RDKit, FlexiFlow achieves comparable results to RDKit in terms of AMR and Cov, with a slight improvement. Specifically, on Coverage recall we can note that with a threshold of 0.75 Å FlexiFlow already achieves a Cov of 1.0, meaning that for all the molecules generated we can find a conformer that is at most 0.75 Å away from a CREST-reference conformer. This suggests that FlexiFlow learns the conformational distribution of the training data and is able to generate conformers that closely resemble the CREST-reference conformers. Conversely, coverage precision decreases because FlexiFlow deliberately explores a broader conformational space, causing some samples to lie farther from the CREST references. This trade-off is expected for a diversity-oriented generator. Overall, FlexiFlow yields high-quality, physically plausible conformers that remain close to reference structures while retaining diversity.

**C.4. GEOM Drugs Stratified RMSD Distribution Before and After Energy Minimization**

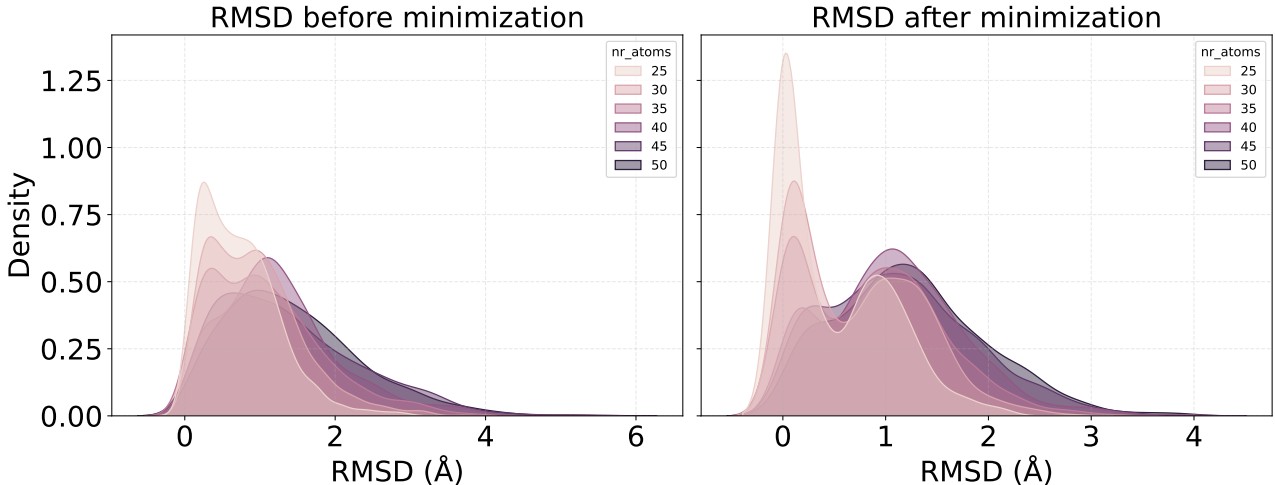

*Figure 10.* Results on GEOM Drugs. We use the metric in Equation 14 to test whether generated conformers, after minimization, occupy distinct local minima of the potential energy surface. For each atom-count setting, we sample 20 molecules; each yields 50 conformers. For every molecule we record the minimum pairwise RMSD (closest conformers). We then minimize each conformer for 100 steps with the MMFF94 force field (Halgren, 1996) and recompute the minimum RMSD. Results show that FlexiFlow's conformers, once minimized, fall into different local minima, indicating captured conformational diversity. The effect strengthens with molecular size: larger molecules exhibit larger pre/post minimization closest-conformer RMSD gaps.

## C.5. QM9 Ablation Comparing Different Model Settings

| Ms-Ep-NFE | Valid ↑ | Novel ↑ | Strain x ↓ | Strain y ↓ |
|---|---|---|---|---|
| S-10-50 | $96.9_{\pm0.16}$ | $99.9_{\pm0.01}$ | $6.13_{\pm0.02}$ | $1.51_{\pm0.02}$ |
| S-10-100 | $95.8_{\pm0.15}$ | $96.4_{\pm0.07}$ | $5.63_{\pm0.01}$ | $1.43_{\pm0.02}$ |
| S-10-500 | $94.9_{\pm0.17}$ | $95.7_{\pm0.08}$ | $5.11_{\pm0.07}$ | $1.53_{\pm0.05}$ |
| S-15-50 | $97.8_{\pm0.10}$ | $99.9_{\pm0.01}$ | $5.95_{\pm0.04}$ | $2.41_{\pm0.02}$ |
| S-15-100 | $95.9_{\pm0.13}$ | $100.0_{\pm0.00}$ | $5.40_{\pm0.01}$ | $2.05_{\pm0.03}$ |
| S-15-500 | $95.8_{\pm0.13}$ | $100.0_{\pm0.00}$ | $4.98_{\pm0.02}$ | $2.39_{\pm0.06}$ |
| M-10-50 | $97.0_{\pm0.12}$ | $93.9_{\pm0.06}$ | $4.03_{\pm0.01}$ | $2.80_{\pm0.06}$ |
| M-10-100 | $97.2_{\pm0.15}$ | $99.1_{\pm0.04}$ | $3.54_{\pm0.04}$ | $1.93_{\pm0.02}$ |
| M-10-500 | $98.3_{\pm0.13}$ | $97.1_{\pm0.06}$ | $3.34_{\pm0.06}$ | $1.03_{\pm0.02}$ |
| M-15-50 | $98.0_{\pm0.13}$ | $91.9_{\pm0.03}$ | $3.54_{\pm0.04}$ | $1.75_{\pm0.02}$ |
| M-15-100 | $100.0_{\pm0.00}$ | $95.8_{\pm0.07}$ | $3.18_{\pm0.03}$ | $1.31_{\pm0.04}$ |
| M-15-500 | $100.0_{\pm0.00}$ | $96.7_{\pm0.07}$ | $3.02_{\pm0.01}$ | $1.82_{\pm0.03}$ |
| L-10-100 | $99.9_{\pm0.01}$ | $91.8_{\pm0.03}$ | $2.12_{\pm0.03}$ | $0.88_{\pm0.04}$ |
| L-20-100 | $99.9_{\pm0.01}$ | $89.9_{\pm0.01}$ | $1.11_{\pm0.05}$ | $0.61_{\pm0.02}$ |
| L-30-100 | $99.9_{\pm0.01}$ | $90.7_{\pm0.01}$ | $0.69_{\pm0.01}$ | $0.48_{\pm0.02}$ |
| L-40-100 | $99.9_{\pm0.01}$ | $90.0_{\pm0.01}$ | $0.51_{\pm0.01}$ | $0.24_{\pm0.01}$ |

*Table 4.* The table shows additional metrics on QM9. Model size (Ms), S (17.2M), M (24.7M) and L (37.7M) params (see Appendix A.6 for details), Epochs (Ep) and number of inference steps (NFE). For all runs reported the Uniqueness is $100.0_{\pm0.0}$. The strain of the top-10 $y$ conformers for each $x$ generated molecular graph is reported.

## C.6. Qualitative Results Conformers GEOM Drugs

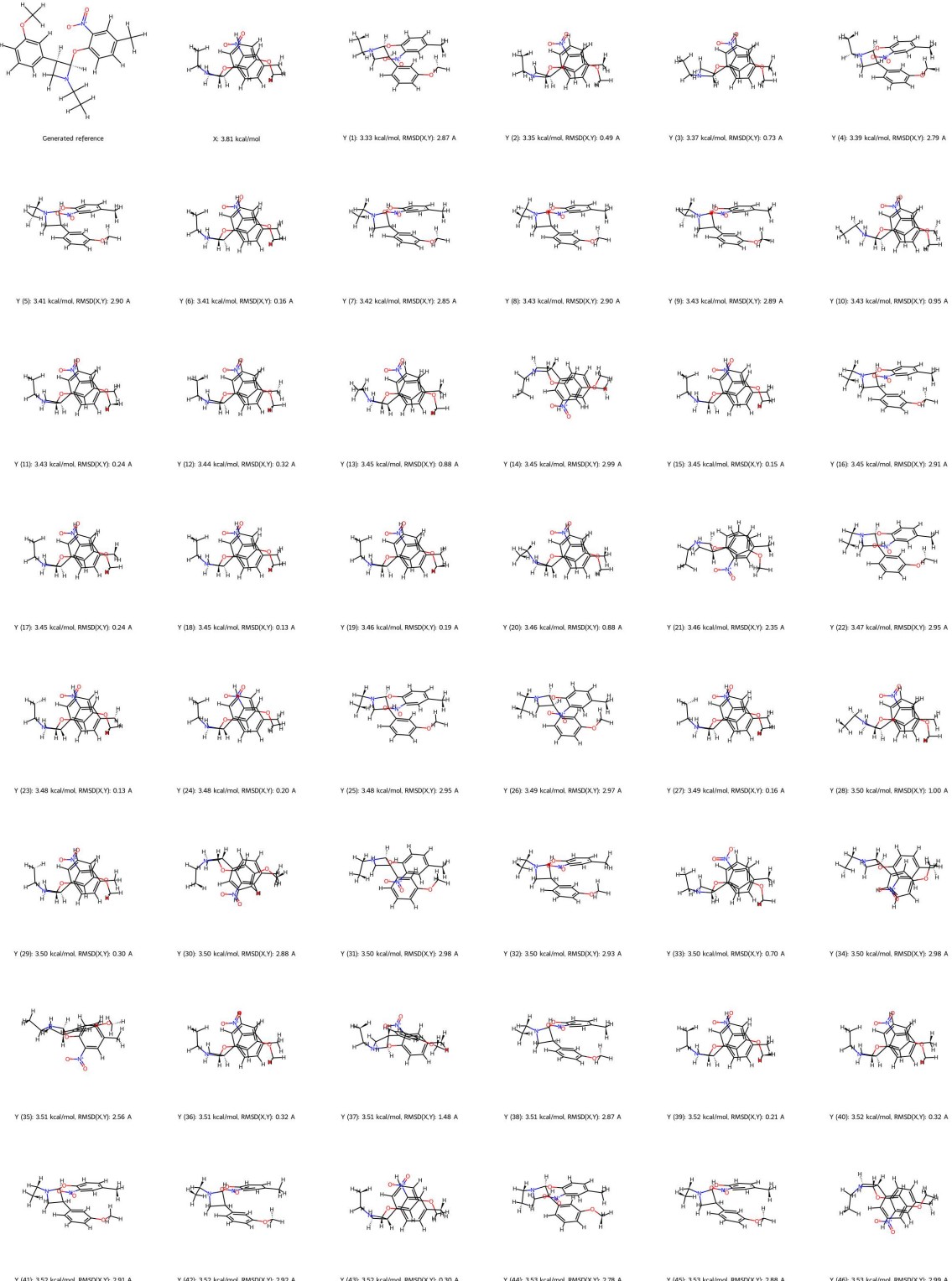

*Figure 11.* Qualitative results of a molecule sampled with FlexiFlow along with its generated conformers. We denote with $x$ the target conformation, and with $y_1, y_2, ..., y_{46}$ the generated conformers. For each molecule we report the energy state and for each $y_i$ the RMSD with respect to $x$.

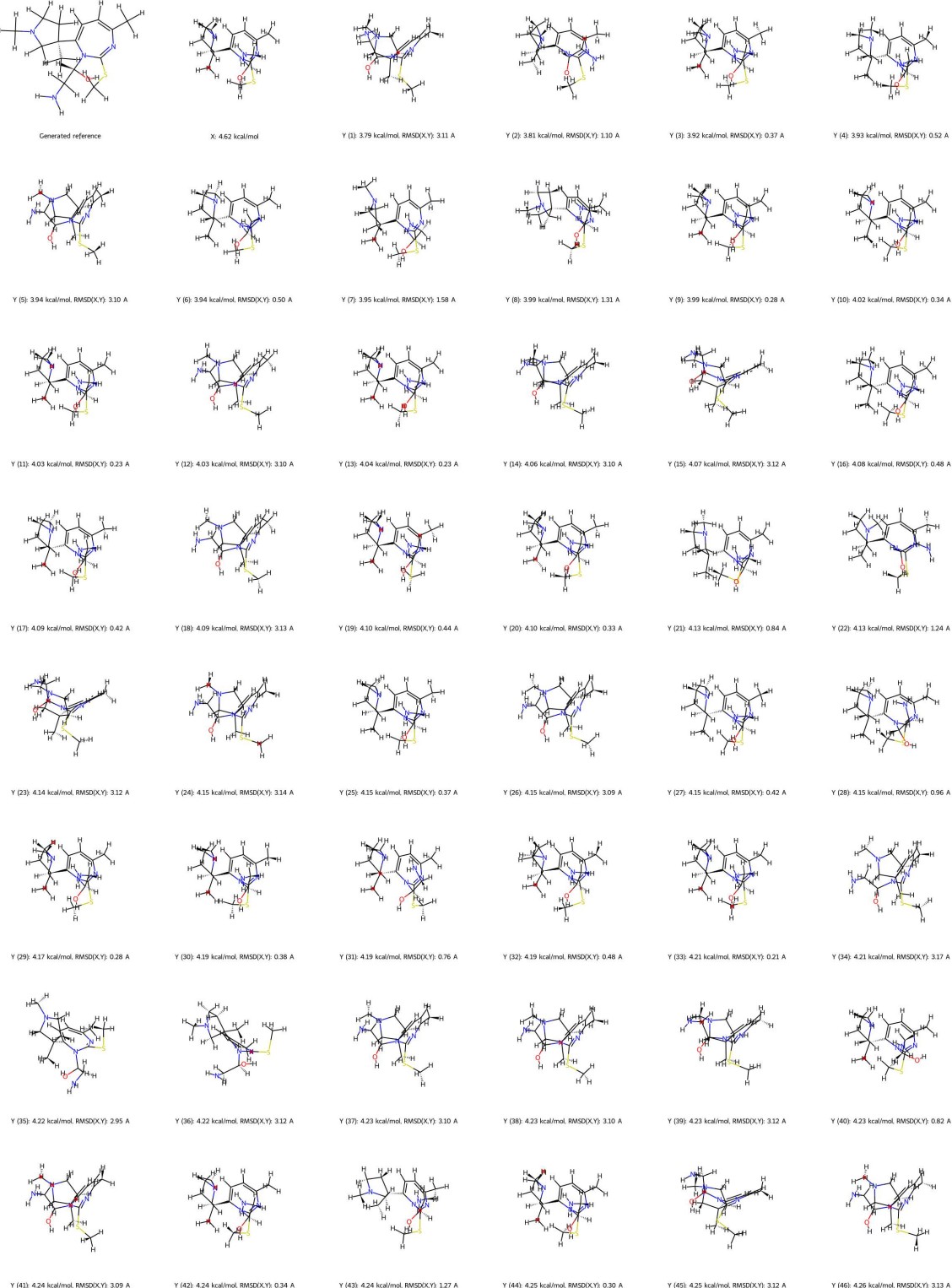

*Figure 12.* Qualitative results of a molecule sampled with FlexiFlow along with its generated conformers. We denote with $x$ the target conformation, and with $y_1$, $y_2$, ..., $y_{46}$ the generated conformers. For each molecule we report the energy state and for each $y_i$ the RMSD with respect to $x$.

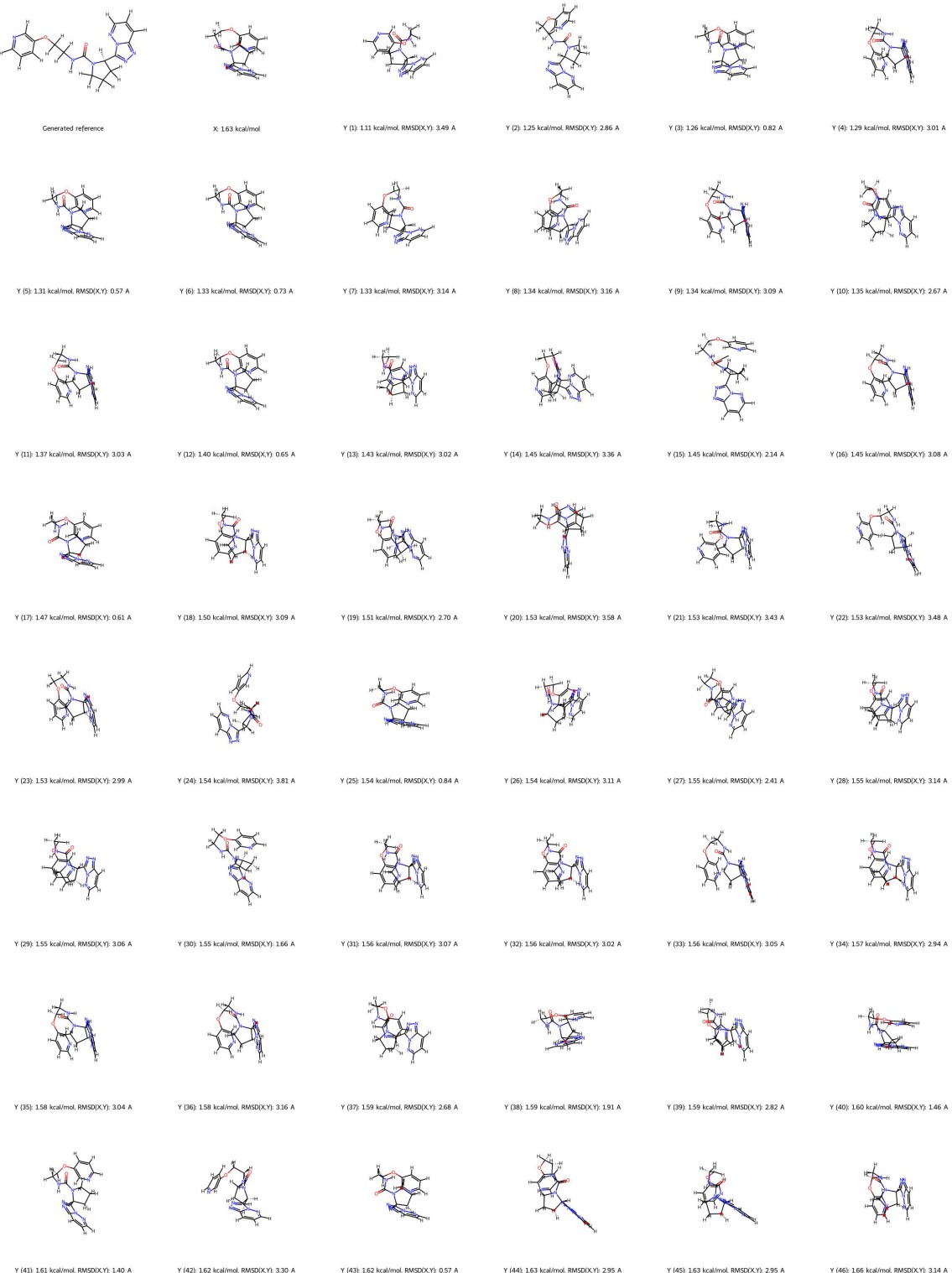

*Figure 13.* Qualitative results of a molecule sampled with FlexiFlow along with its generated conformers. We denote with $x$ the target conformation, and with $y_1$, $y_2$, ..., $y_{46}$ the generated conformers. For each molecule we report the energy state and for each $y_i$ the RMSD with respect to $x$.

## C.7. Protein Conditioning with and without Molecular Flexibility During Training

For this experiment, we generate 10 $x$ molecular structures along with 42 $y$ conformers each for each testset protein in PDBBind using both models with and without GEOM Drugs training. By performing inference on a consumer GPU (RTX 3060 with 6GB of vRAM), we could generate only 42 $y$ conformers to avoid out of memory issues when conditioning on larger pockets. However, in this case the implementation is not fully optimized, as it recompute the reference molecule features for each conformer $y$. Computing the reference molecule features once would significantly improve the efficiency. We compute the MMFF strain energy for all generated conformers as the difference between the MMFF energy before and after MMFF optimization, divided by the number of atoms in the molecule. Training the model with GEOM Drugs consistently yields lower and tighter MMFF strain energy distributions than the training without it. The violin plots in Figure 14 illustrate this trend across different top-$k$ $y$ conformers, where the average is performed on the MMFF strain. We better quantify the gap (in terms of MMFF strain between the model with and without GEOM Drugs training) in Figure 15, where we show the difference between the two models' mean MMFF strain.

*Figure 14.* Violin plots report the mean MMFF strain (per-atom conformational strain = (unoptimized MMFF energy - optimized MMFF energy) / atom count) for the top-$k$ $y$ ligand conformations ($k = 1, 5, 10, 15, 20, 30$) with or without GEOM Drugs training. Boxes inside violins mark mean and interquartile range. The log y-axis highlights a heavy right tail, without GEOM Drugs exhibit progressively higher mean strain as $k$ increases, while with GEOM Drugs maintains lower and more compact distributions across all $k$. This indicates including the flexibility during training (from GEOM Drugs) consistently produces lower-strain (more relaxed) top-ranked conformations. The magnitude growth of this advantage is quantified in the accompanying Delta-strain plot (Figure 15).

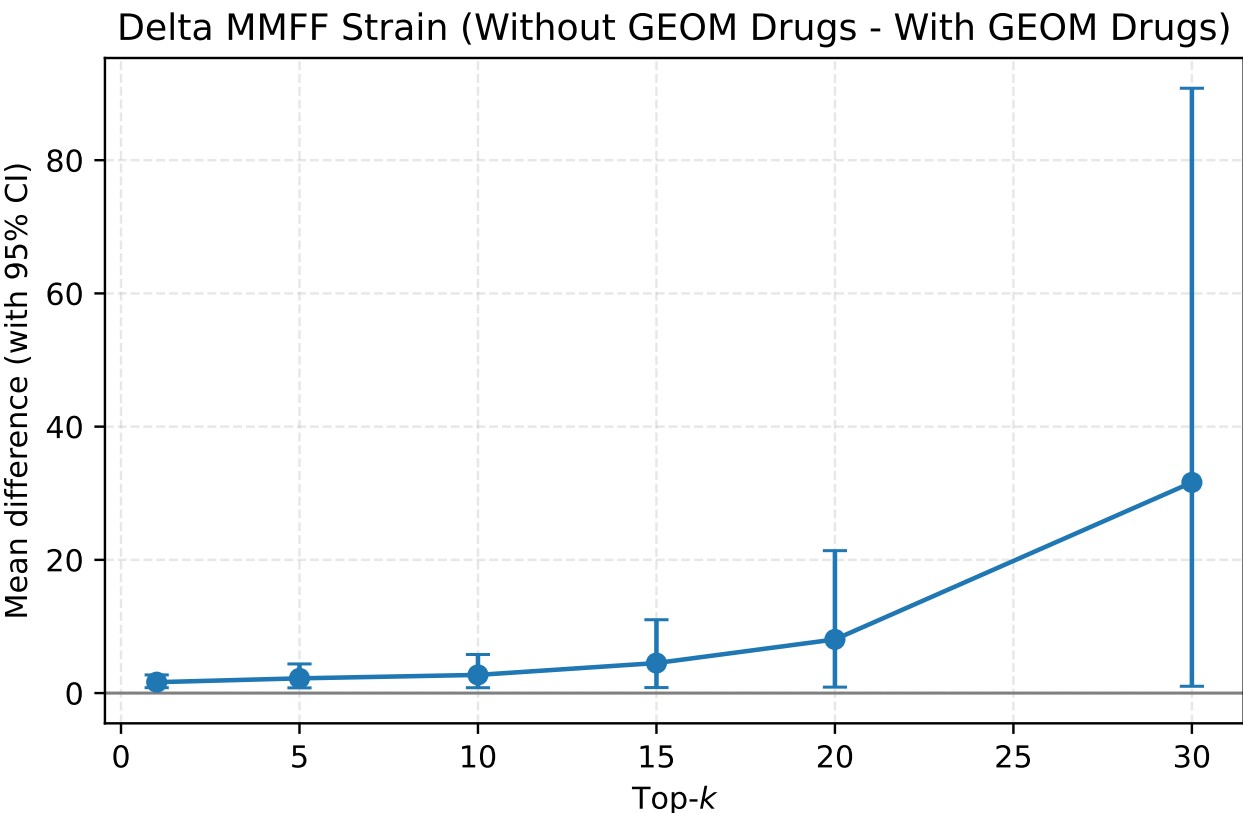

*Figure 15.* Figure reports the mean difference in MMFF strain between without and with GEOM Drugs during training for the top-$k$ $y$ ligand conformations ($k$ = 1, 5, 10, 15, 20, 30) with 95% confidence intervals (error bars).

## C.8. Additional Results on Protein Conditioning

Although FlexiFlow is neither trained nor explicitly conditioned to optimize Vina scores on the conformer coordinates $y$, some generated conformers nevertheless obtain higher Vina scores. This implies the model implicitly captures protein-ligand interaction geometry. Figures 16, 17, 18 and 19 (complex 6e5s) illustrate a qualitative example: conditioned on the protein pocket, we generate the target ligand $x$ and multiple conformers $y$; three $y$ conformers are shown alongside the reference ligand pose.

Additionally, we provide additional illustrations of other generated ligands on the testset including both target $x$ and conformer $y$, see Figures 20, 21 and 22.

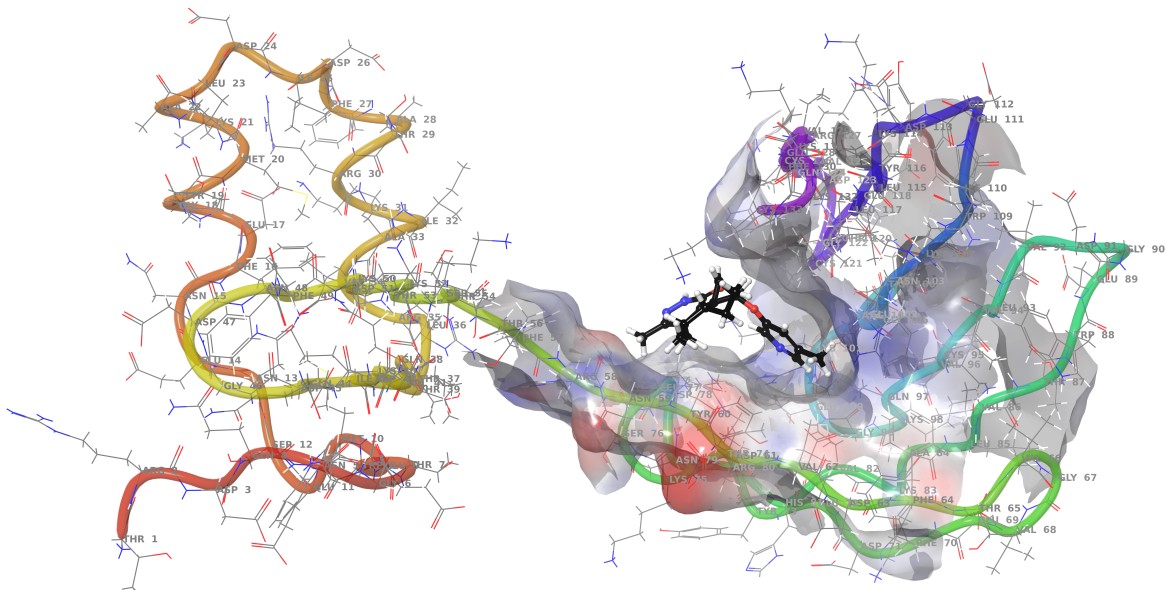

*Figure 16.* Binding surface electrostatic potential for complex 6e5s mapped onto the ligand's generated reference conformation $x$ (Vina score: -4.1 kcal/mol; QED: 0.94).

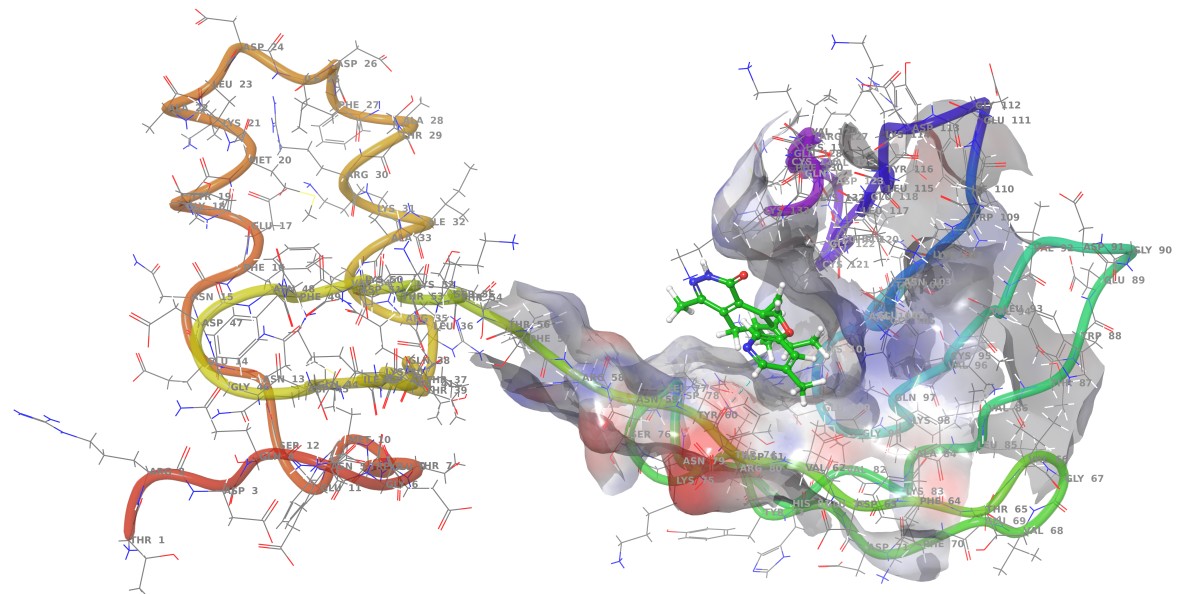

*Figure 17.* Binding surface electrostatic potential for complex 6e5s mapped onto the ligand's generated reference conformation $y_1$ (Vina score: -5.5 kcal/mol; QED: 0.94).

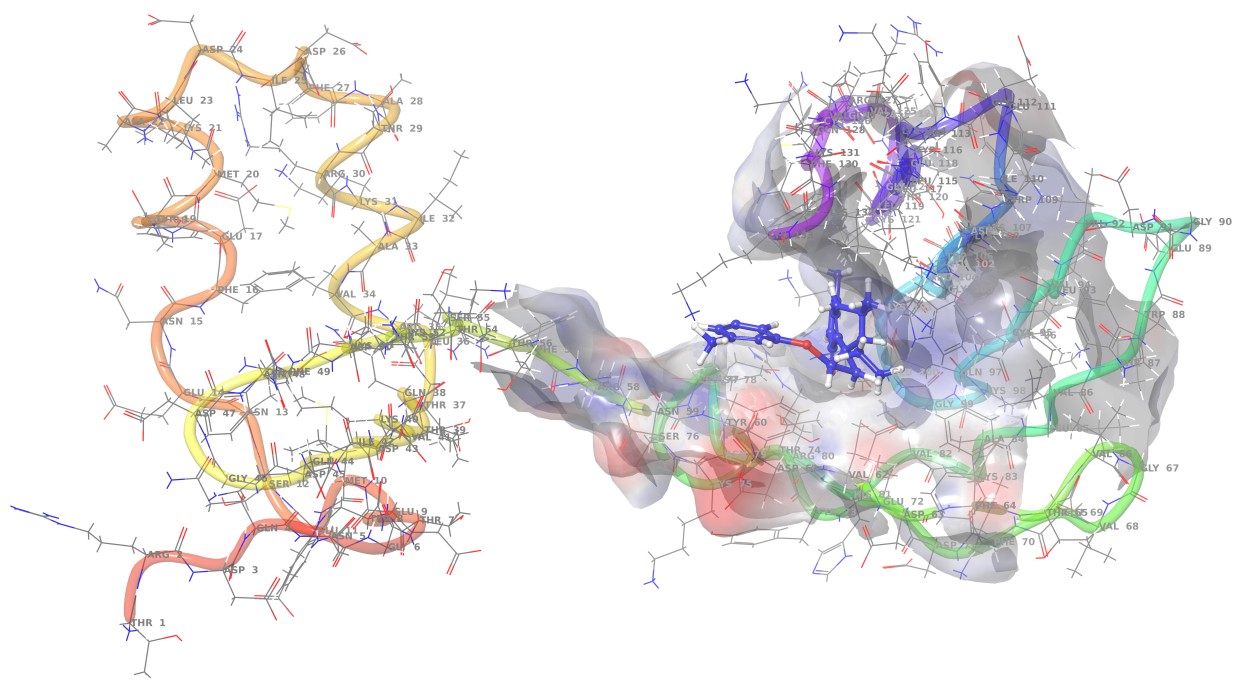

*Figure 18.* Binding surface electrostatic potential for complex 6e5s mapped onto the ligand's generated reference conformation $y_2$ (Vina score: -5.3 kcal/mol; QED: 0.94).

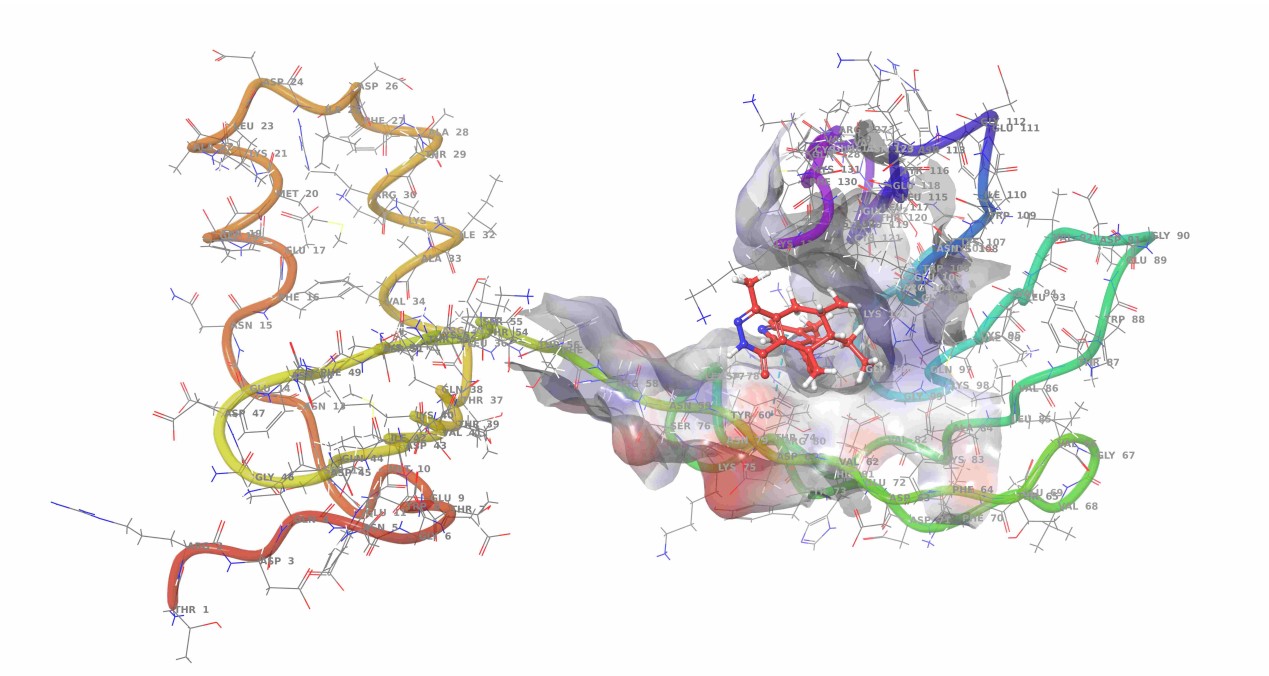

*Figure 19.* Binding surface electrostatic potential for complex 6e5s mapped onto the ligand's generated reference conformation $y_3$ (Vina score: -5.0 kcal/mol; QED: 0.94).

## C.9. Qualitative Results on Protein Conditioning x and y Conformations

.

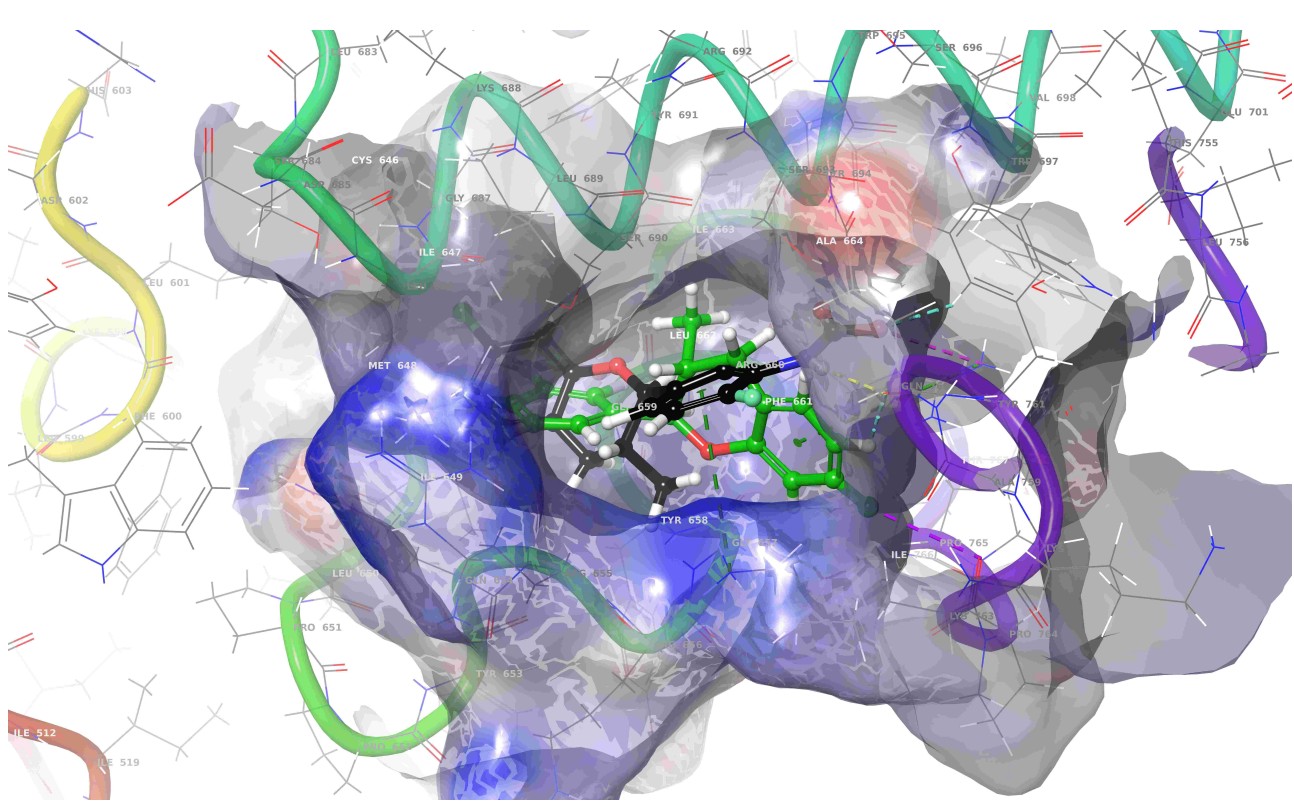

*Figure 20.* Conformations $x$ (Vina score: -3.1 kcal/mol) and $y$ (Vina score: -6.0 kcal/mol) for the complex 6oin (QED ligand: 0.91).

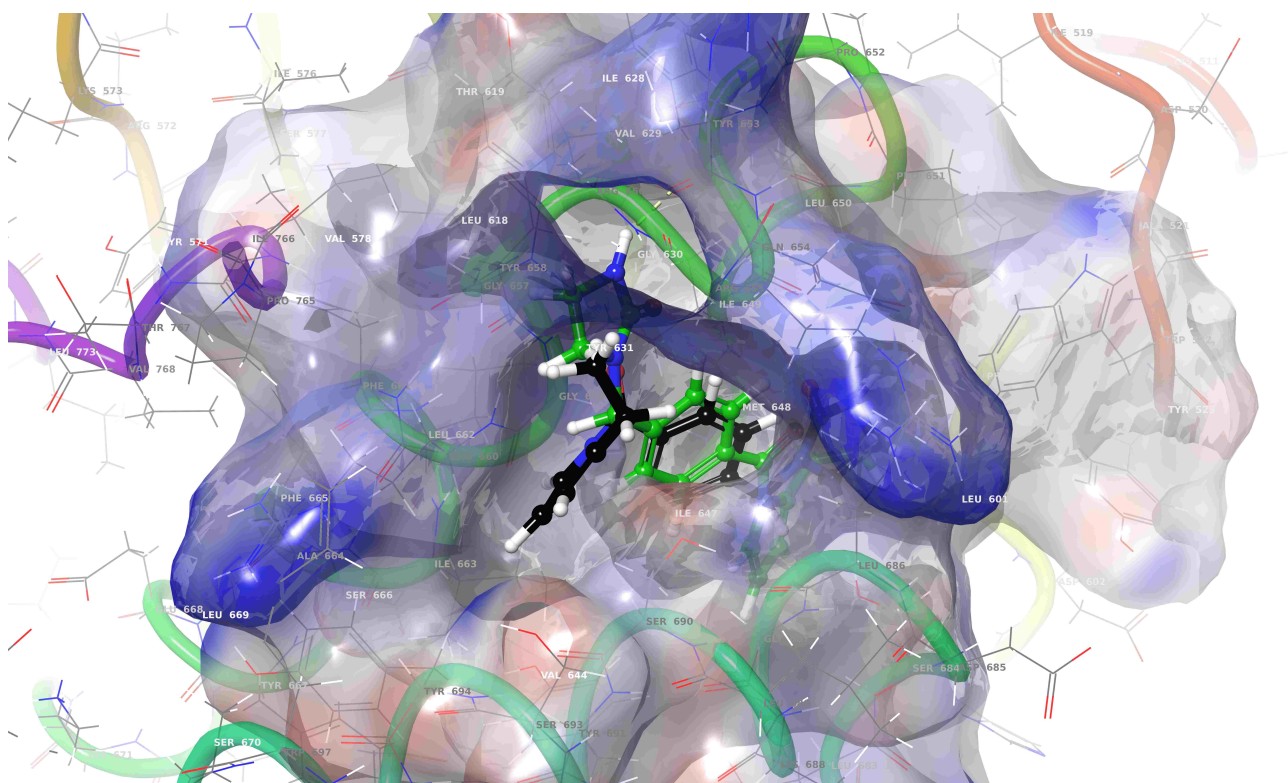

*Figure 21.* Conformations $x$ (Vina score: -2.3 kcal/mol) and $y$ (Vina score: -4.8 kcal/mol) for the complex 6oiq (QED ligand: 0.94).

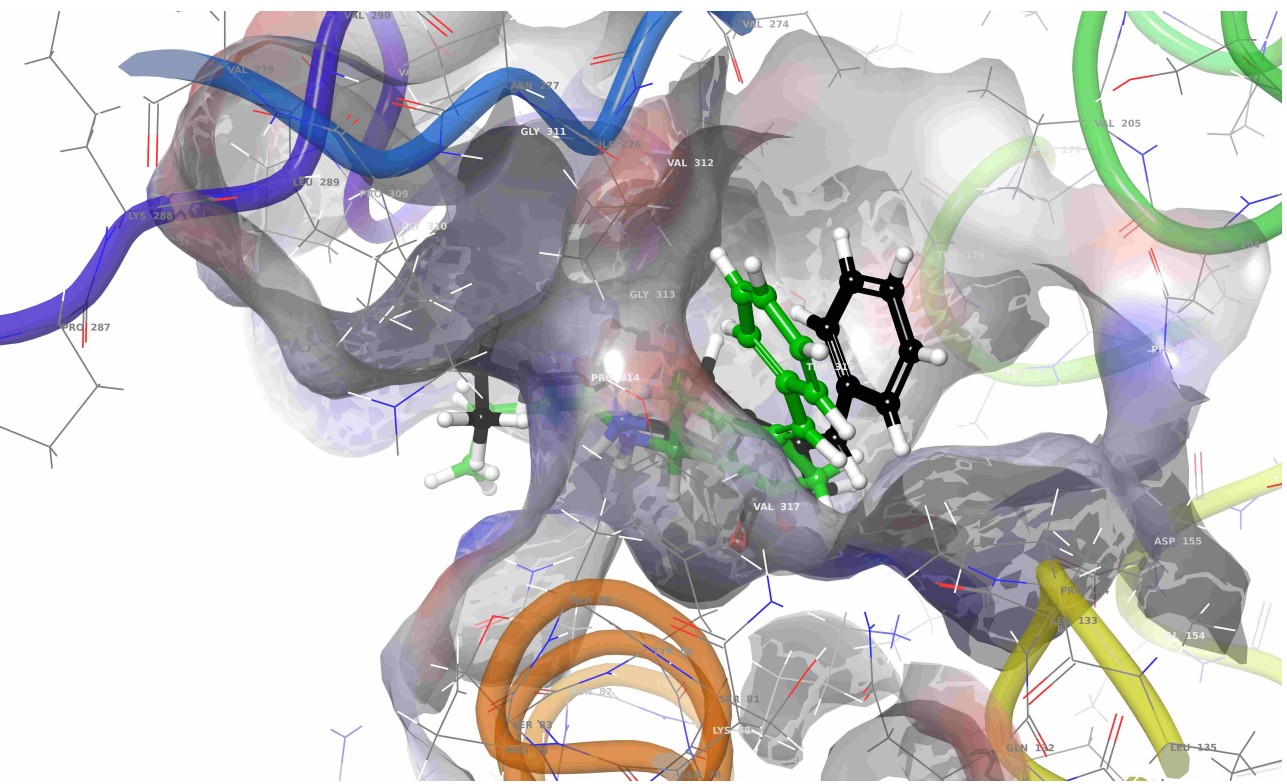

*Figure 22.* Conformations $x$ (Vina score: -4.5 kcal/mol) and $y$ (Vina score: -4.6 kcal/mol) for the complex 6jib (QED ligand: 0.92).

## D. Flow Decomposition on MNIST

### D.1. Data Processing & Training Setup

Since MNIST does not have implicitly colored images for the digits, we color the digits using the following procedure. First, we uniformly sample a color among red, green and blue and added some white noise over it. Then, this is multiplied by the grayscale value of the digit, so that the background remains black.

In order to independently sample the color based on the digit $x$ that is being generated, we use the sum between the grayscale digit $x$ and RGB color $y$ to constrain the generation of a colored digit $z$, i.e. $z = x + y$. In this way, the model can learn to generate both the grayscale digit and the color independently (see Figure 23).

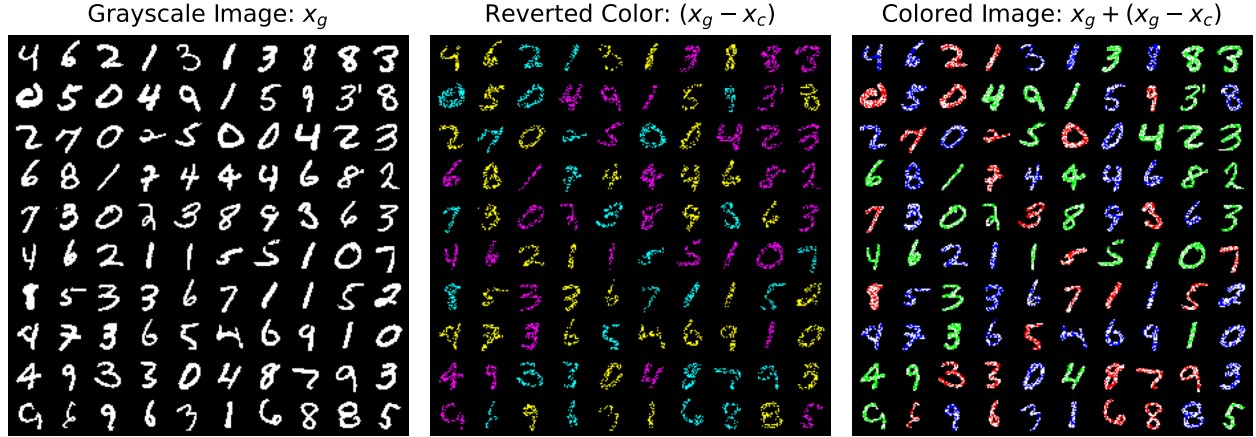

*Figure 23.* Data processing steps used to derive the colored digit $x + y$ from the grayscale digit $x = x_g$ and the $y = x_g - x_c$ vector field.

### D.2. Dual-Unet MNIST Architecture

The Dual-Unet architecture $\mathcal{F}_{\text{Dual-Unet}}(x_t^g, x_t^y, t) = (s_g, s_c)$ is a modified U-Net that takes as input the noisy grayscale image $x_t^g \in \mathbb{R}^{1 \times H \times W}$, and noisy reversed color image $x_t^y \in \mathbb{R}^{3 \times H \times W}$ and time $t$, and outputs the score functions $s_g$ and $s_y$ for the grayscale and reversed color images (rescaled by a factor $0.001$). In this setting, we train the network to directly estimate the target data distribution from the noisy inputs.

To obtain $x_t^g$ and $x_t^y$, we sample a $x_0^g$ and $x_0^y$ independently from a $\mathcal{N}(0, I)$ distribution, $t \sim \text{Uniform}(0, 1)$ and we use the following interpolation scheme: $x_t^g = (1 - t)x_0^g + tx_1^g$ and $x_t^y = (1 - t)x_0^y + tx_1^y$, where $x_1^g$ and $x_1^y$ are the grayscale and reversed colored images from the training set, respectively. During training we minimize the objective in Equation 9.

In the architecture, $x_t^g$ and $x_t^y$ are flatten and process by $x_{g,0} = \text{Conv}_g(x_t^g) \in \mathbb{R}^{D \times H \times W}$, for the colored counter part $x_{y,0} = \text{Conv}_c(x_t^y) \in \mathbb{R}^{3D \times H \times W}$ and $\tau(t) = \text{MLP}(\text{SinusoidalEmbed}(t)) \in \mathbb{R}^{4D}$ for the time component, where Conv stands for convolutional layer. We apply $L$ encoder layers $\mathbf{E}_i, i \in 1, \ldots, L$:

$$h_{g,i}, x_{g,i} = \mathbf{E}_{g,i}(x_{g,(i-1)}, \tau) \quad h_{y,i}, x_{y,i} = \mathbf{E}_{y,i}(x_{y,(i-1)}, \tau) \tag{77}$$

where $\mathbf{E}_{g,i}$ and for the colored $\mathbf{E}_{y,i}$ are encoder blocks with ResNet and downsampling blocks, producing features $x_{g,L}$, $x_{y,L}$ and skip connections $h_{g,L}, h_{y,L}$. We finally concatenate the colored to the grayscale bottleneck features and from these start decoding the images $x_{g,m} = \mathcal{M}_g(x_{g,L}, \tau)$ and $x_{y,m} = \mathcal{M}_c([x_{y,L}, x_{g,m}], \tau)$ where $[\cdot, \cdot]$ denotes channel-wise concatenation. We reverse the process with a decoder $\mathbf{D}_i$ for $i \in L, \ldots, 1$ layers for both $x_{g,m}$ and $x_{y,m}$:

$$x_i^u = \mathbf{D}_{g,i}([x_{g,(i+1)}^u, h_{g,i}], \tau) \quad x_{y,i}^u = \mathbf{D}_{y,i}([x_{y,(i+1)}^u, h_{y,i}, h_{g,i}], \tau) \tag{78}$$

where $\mathbf{D}_{g,i}$ and $\mathbf{D}_{y,i}$ are decoder blocks with ResNet and upsampling blocks. The output is the result of a ResBlock and final convolution

$$s_g = \text{Conv-l}_g(\text{ResBlock}_g([x_{g,1}^u, x_{g,0}], \tau)), \ s_c = \text{Conv-l}_y(\text{ResBlock}_y([x_{y,1}^u, x_{y,0}, x_{g,0}], \tau)) \tag{79}$$

### D.3. MNIST Inference

At inference time, we sample $\hat{x}_0^g$ and $\hat{x}_0^y$ independently from a Gaussian distribution $\mathcal{N}(0, I)$, and we use the euler ODE solver to integrate both vector fields from $t = 0$ to $t = 1$ independently. Finally, we obtain the grayscale image $\hat{x}_1^g$ and reversed color image $\hat{x}_1^c$, to obtain the final colored image we simply apply the operation we show in Figure 23, $\hat{x}_1^g + (\hat{x}_1^g - \hat{x}_1^y)$.

### D.4. Qualitative Results on MNIST

For Figure 24, the noise for $x_g$ is held fixed while varying the noise for $x_c$, producing distinct color textures conditioned on an unchanged grayscale digit structure. Figure 25 provides results varying the noise for both $x_g$ and $x_c$ yields diverse grayscale digit shapes together with corresponding variations in color texture. This demonstrates that FlexiFlow can independently modulate structure (grayscale) and appearance (color or texture).

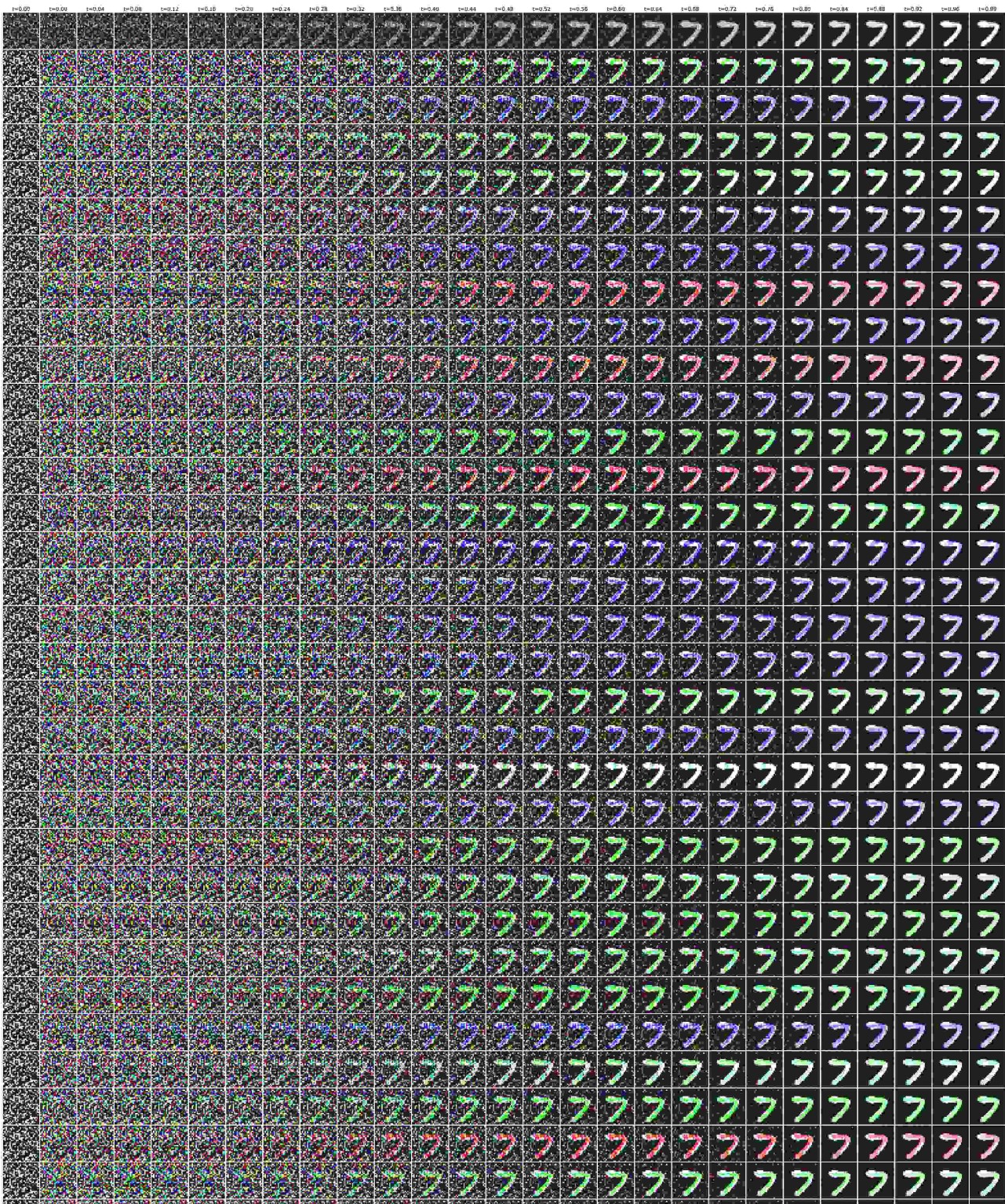

*Figure 24.* Integration of different $x_c$ noise for a fixed $x_g$ noise, where we can observe that FlexiFlow is able to generate different color textures for the same grayscale image. The $x_g$ reference is shown in the top row, while all the other rows show different $x_c$ samples.

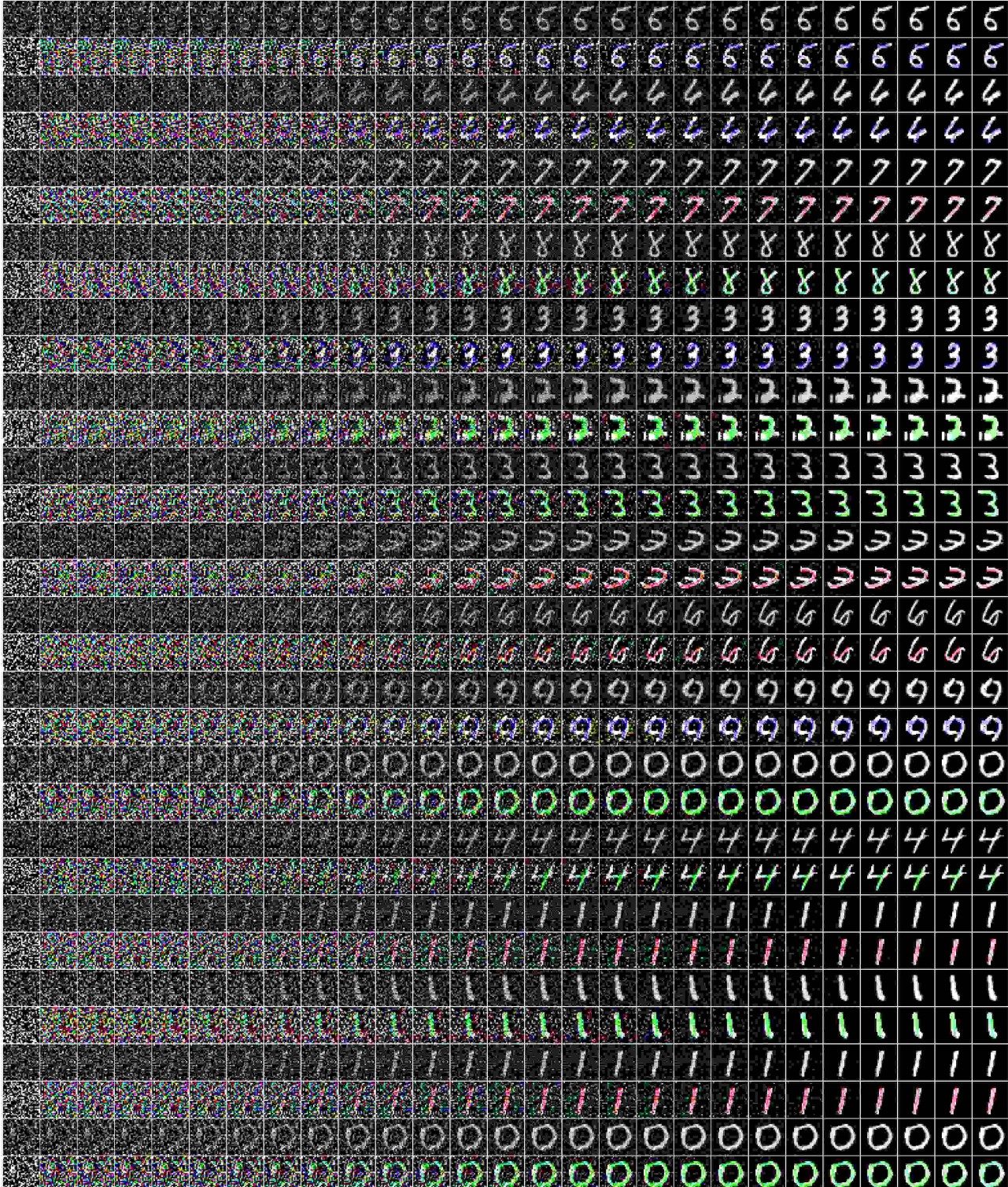

*Figure 25.* Integration of different $x_g$ and $x_c$ noise, where we can observe that FlexiFlow is able to generate diverse grayscale images along with different color textures. The $x_g$ reference is shown every other row, while all the other rows show different $x_c$ samples given the same $x_g$ noise shown on top.

## E. Training Dataset Energies Statistics on GEOM Drugs Constructed (x, y) Pairs

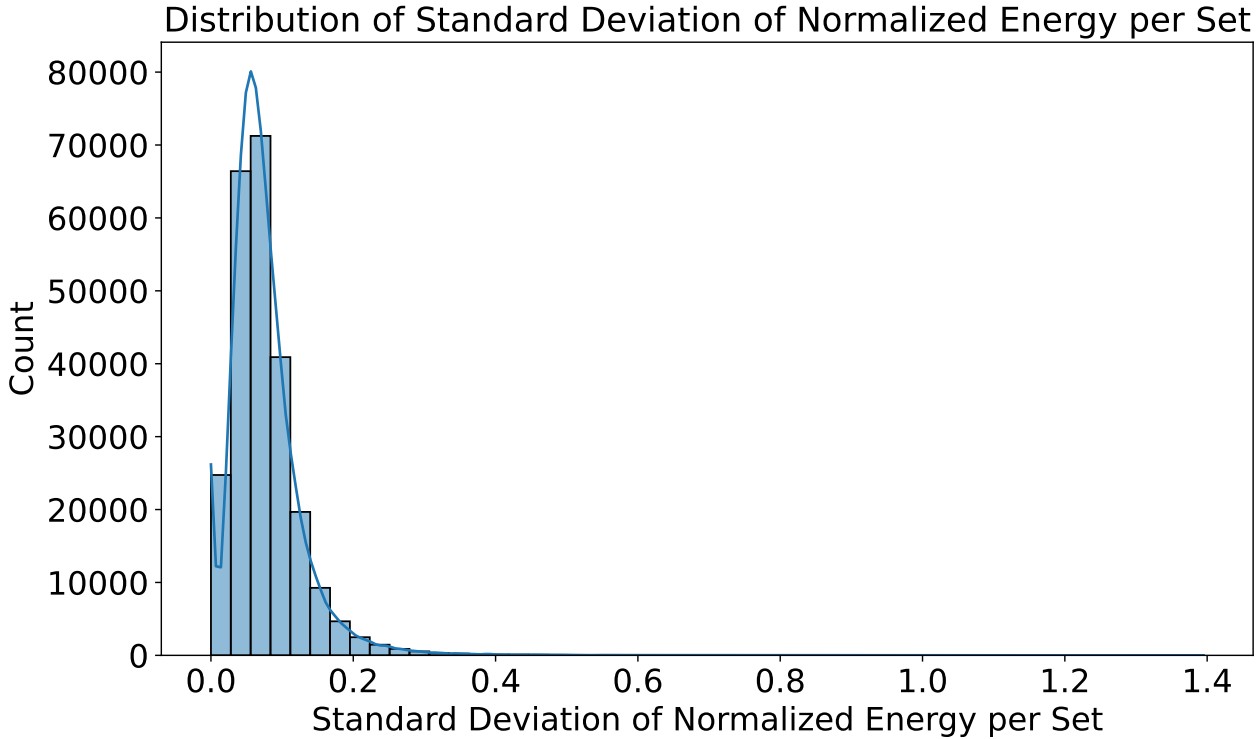

*Figure 26.* Standard deviations of the normalized energies (computed as described in Appendix C.1) of the molecules within each set of conformers $\mathcal{S}$. This plot shows that all the conformations that we have in GEOM Drugs, in terms of energy, are almost at the energy minima configuration.

| Energy Minima Location | Count |
|:---:|:---:|
| $y$ | 224265 |
| $x$ | 19425 |

*Table 5.* Count of how many times the energy minima conformation is located in $x$ or $y$ by selecting the closest conformer to the average as reference.

| Energy difference | Count |
|:---:|:---:|
| $y < x$ | 224265 |
| $y > x$ | 15378 |
| $x = y$ | 4047 |

*Table 6.* Times the energy minima conformation in $x$ or $y$ is lower, or equal, by selecting the closest conformer to the average as reference.

## F. Inference Overhead FlexiFlow Compared to SemlaFlow

| Method | Total time (s) | # Generated Conformers | # Generated Graphs | Inference Time per Conformer |
|---|---|---|---|---|
| SemlaFlow | 142.3862 | 1000 | 1000 | 0.1423862 |
| FlexiFlow | 185.3495 | 2000 | 1000 | 0.09267475 |

*Table 7.* Computation time per conformer for SemlaFlow and FlexiFlow.

## G. Additional Metrics GEOM Drugs

| Model | %PDB Valid ↑ | OOD Ring ↓ | FG dev ↓ |
|---|---|---|---|
| FlowMol3 | 91.9±0.7 | 0.27±0.03 | 0.10±0.01 |
| FlexiFlow | 91.3±0.3 | 0.38±0.02 | 0.27±0.01 |
| SemlaFlow | 88.5±1.3 | 0.35±0.02 | 0.20±0.02 |
| EquiGAT-diff | 77.6±0.8 | 0.58±0.03 | 0.28±0.01 |

*Table 8.* Additional metrics for de novo molecular generation on the GEOM Drugs computed according to the FlowMol3 (Dunn & Koes, 2025).

## H. Comparison Table Different Method

| Method | Generate molecules | Generate conformers | Speed |
|---|---|---|---|
| Boltzmann Generators | ✗ | ✓ | ✗ |
| Adjoint Sampling | ✗ | ✓ | ✓ |
| SemlaFlow | ✓ | ✗ | ✓ |
| FlexiFlow | ✓ | ✓ | ✓ |

*Table 9.* Comparison of molecular generation, molecular conformation generation, boltzmann generators and FlexiFlow.

