# OpenReview forum: "FlexiFlow: decomposable flow matching for generation of flexible molecular ensemble"
_ICML.cc/2026/Conference — ICML 2026 regular_

### Official Review · Reviewer_sY2T · 2026-02-26

**Soundness:** 2
**Presentation:** 3
**Significance:** 2
**Originality:** 3
**Overall Recommendation:** 4
**Confidence:** 2

**Summary:**

FlexiFlow proposes a flow-matching based architecture that can jointly generate molecular graphs and multiple conformations, supporting both unconditional and conditional generation. The method is evaluated on QM9 and GEOM-Drugs for molecular generation quality as well as conformer quality/diversity, and is further applied to an SBDD-style task.

**Compliance With Llm Reviewing Policy:**

Affirmed.

**Final Justification:**

My main concern centered on the scientific significance of multi-conformer generation without guarantees of sampling the full Boltzmann distribution. I accept the authors’ explanations, so I have raised my score to 4. I encourage the authors to incorporate this discussion on the method’s significance and limitations into the revised manuscript.

**Key Questions For Authors:**

1. Please clarify the practical value of the proposed method for real scientific/industrial applications. For example, what do the authors see as a plausible path from their model to the estimation of thermodynamic quantities such as ΔG, binding free energies, or conformational entropies?

2.In Tables 1 and 2, are the reported results based on a single generated conformer per molecule, or do you generate multiple conformers and report the best metric among them? Why does FlexiFlow outperform SemlaFlow, could this be largely due to training on more conformer data? In Table 2, atom/molecule stability improves but validity decreases; Could you please add an explanation or analysis for this "abnormal" result?

3. I recommend adding a schematic or a more explicit comparison between FlexiFlow and Adjoint Sampling / Boltzmann Generators, highlighting the differences in problem setting and methods

**Limitations:**

Eq.2 relies on a conditional independence assumption. Could the authors discuss its limitations?

**Strengths And Weaknesses:**

Strengths：
- Jointly generating molecular graphs together with conformers sampled according (ideally) to a Boltzmann-like distribution is an important problem for the community.
- The proposed architecture and flow-matching decomposition are a reasonable and interesting attempt in this direction.
- The experimental section is extensive and shows strong, often SOTA-level performance on standard benchmarks.

Weaknesses：

1. Lack scientific significance demosrtation. The multi-conformer generation remains largely at the geometric level: the model does not attempt to learn or assess approximate Boltzmann weights, nor does it use the generated ensembles to estimate thermodynamic quantities such as ΔG, binding free energy, or conformational entropy. The motivation at the scientific problem level is underdeveloped. Moreover, the authors themselves note that many conformers in Geom are very similar; training multi-conformer generation on such data likely focuses on fitting within a single energy basin, and it is unclear how much the model learns about cross-basin transitions or binding-relevant large-scale flexibility.

2.The PDBBind evaluation lacks quantitative comparison to baselines. Besides, it would be helpful to analyze whether certain pocket types or ligand flexibility regimes benefit more from the proposed multi-conformer capability.

3. Issues with molecule generation benchmark. To my knowledge, in Tables 1 and 2 the “upper” and “lower” groups of methods use different implementations of stability metrics, making them not directly comparable; this is also likely why the reported stability differ so much. In this context, the statement that earlier methods "often produced unstable structures, but newer architectures and training strategies ... have rapidly improved" is somewhat misleading. In addition, several metrics in the lower parts of the tables are essentially saturated, which reduces their usefulness as comparative indicators.

If the authors can provide a more convincing demonstration of the significance and resolve the raised problems, I will reconsider the score.

---

> ### Author Rebuttal · Authors · 2026-03-30
>
> W1.Thank you for the comment. As clarified in the Abstract, Introduction, and Experiments, our method does not explicitly approximate the Boltzmann distribution or estimate Boltzmann weights. While our long-term goal is modeling thermodynamic processes (e.g., binding free energies), several steps are still required. Instead, we present a simple extension of flow-matching molecule generators to produce ensembles of minima. Compared to exhaustive physics-based sampling with CREST, our model achieves high recall and precision, producing conformers comparable in energy and coverage, meaning that the conformers produced by our method are approximately as good as those found by CREST in terms of energy and exhaustiveness of sampling. These can support thermodynamic estimates similarly to CREST when reweighted with a chosen potential. However, limitations arise from the training data: as a data-driven approach, sampling fidelity depends on GEOM Drugs, where conformers are generated using GFN2-xTB in CREST. Moreover, since druglike molecules are typically small and rigid, the model is unlikely to generalize to larger, more flexible systems such as binding pockets.
>
> W2. Thank you for the suggestion. As this is a new evaluation setting, establishing a fair quantitative comparison to existing baselines is non-trivial. The core idea of this experiment is to provide evidences that multi-conformer training can help to explore multiple poses as well. We agree that analyzing performance across pocket types and ligand flexibility regimes is important, and we are actively exploring this direction. However, it requires jointly modeling ligand and pocket flexibility, which entails substantial architectural and methodological changes and is beyond the scope of the current work.
>
> W3. Thank you for the feedback. We will revise the potentially misleading statement, clarify differences in metric implementations, and note that results are not directly comparable. While atom/molecular stability is reported, it is less informative than energy. We also acknowledge that some metrics are saturated (and include additional ones as suggested by another reviewer). Our goal is not to improve these metrics, but to match performance while tackling the more challenging task of generating multiple conformers per molecule. We have provided additional details in Q2.
>
> Q1. As eluded to above, the current approach does not provide direct access to quantities such as binding energies. However, we do believe that co-producing molecules and representative local minima has value in and of itself, as evidenced by the existence of multiple workflows (CREST) and commercial products (OpenEye Omega) for exactly this purpose. For one thing, such ensembles provide can be used to estimate molecular-level properties such sigma distributions for solvation free energies (e.g. COMSOconf) or estimating ligand crystal phases. From a drug discovery perspective, understanding the shape of low-lying conformers underpins shape/pharmacophore-based virtual screening, e.g. ROCs - the key idea is to find molecules that can access a bio-active shape/pharmacophore without straining from the ligand distribution. It also serves as basis for estimating strain in docked poses.
> Finally, as we observed in our pocked-conditioned experiment, our approach seems to improve single-conformer results like docking scores sometimes (Fig.6) or strain energies (Appendix C.7). Work is ongoing in our lab to extend this to protein-ligand complex geometries, which is needed to extend to more complex thermodynamic observables.
>
> Q2. Thank you for the questions. We will clarify this in the manuscript. Tables 1 and 2 report metrics on the generated molecular graphs; these are unaffected by generating multiple conformers per molecule. Our goal is to achieve comparable performance while addressing a more challenging task, producing multiple conformers, which can explain the slight drop in validity. We also note that these metrics are largely saturated, so small differences (e.g. in atom/molecule stability) are not necessarily indicative of more useful molecules.
>
> Q3. Thank you for the suggestion. We will add a section in the appendix which includes a schematic that highlights the differences between Adjoint Sampling / Boltzmann Generators / SemlaFlow / FlexiFlow as synthesized in table below.
> |Method|Generate molecules|Generate conformers|Speed|
> |--|--|--|--|
> |Boltzmann Generators|x|✓|x|
> |Adjoint Sampling|x|✓|✓|
> |SemlaFlow|✓|x|✓|
> |FlexiFlow|✓|✓|✓|
>
> L1. Thanks for this comment. We assume conditional independence, as conformers can be viewed as independent phase-space samples—desirable for uncorrelated exploration, unlike molecular dynamics which yields correlated samples. A potential risk is redundant sampling (mode collapse), but we do not observe this empirically; e.g., Figure 3 (left) shows all conformers are distinct with large RMSD.

---

> > ### Author Rebuttal · Reviewer_sY2T · 2026-04-04
> >
> > Thank you for the response. My concerns have been addressed. I look forward to the authors exploring the idea that "supporting thermodynamic estimates similarly to CREST when reweighted with a chosen potential."

---

> > > ### Author Response · Authors · 2026-04-05
> > >
> > > Thank you for the constructive discussion and comments. We really appreciate it. We are happy to see that the rebuttal has addressed your concerns.
> > >
> > > We just wanted to kindly check with you whether, in light of your “fully resolved” assessment, you might consider updating the score, as the current rating seems somewhat misaligned with this evaluation. Of course, we understand if the update is not possible at this stage.
> > >
> > > Thanks again for your time and engagement!

---

### Official Review · Reviewer_JCph · 2026-03-12

**Soundness:** 3
**Presentation:** 2
**Significance:** 3
**Originality:** 2
**Overall Recommendation:** 4
**Confidence:** 3

**Summary:**

The paper introduces FlexiFlow, a generative model based on flow matching that simultaneously generates 3D molecular graphs and multiple corresponding conformations. It directly addresses a key limitation of existing 3D de-novo design models, which typically only produce a single conformation per molecule. Experiments conducted on the QM9 and GEOM Drugs datasets demonstrate state-of-the-art results in 3D molecular generation tasks.

**Compliance With Llm Reviewing Policy:**

Affirmed.

**Final Justification:**

the authors resolve my issues, so I maintain my score at 4.

**Key Questions For Authors:**

- Physical Plausibility of the Conditional Independence Assumption: The paper utilizes a conditional independence assumption in its flow decomposition (Equation 2, assuming each conformer $y_i$ is independent given $x$). However, different low-energy conformations of the same molecule are typically highly correlated within the physical energy landscape. Does this assumption restrict the model's capacity to capture the complex, multi-modal correlations inherent in the true conformational distribution (such as the Boltzmann distribution)? In practical generation, have you observed conformers "collapsing" into similar states, or a lack of specific transition states?
- Consistency between 3D conformations and the 2D molecular graph: FlexiFlow aims to simultaneously generate a molecular graph and multiple 3D conformations. During inference, how does the model strictly guarantee that the multiple generated $y$ conformers (coordinate sets) entirely correspond to the identical 2D molecular graph on a structural parsing level (e.g., when resolving bond lengths and angles)? Are there failure cases where a generated $y$ conformer actually maps to a different molecule due to excessive coordinate deviations (e.g., resulting in unintended isomerization or bond cleavage)?

**Limitations:**

Yes.

**Strengths And Weaknesses:**

Strength:
- Novelty: The joint generation of molecular graphs and multiple low-energy conformations using decomposable flow matching is a novel and highly relevant contribution to computational drug discovery.
- Performance: The model achieves state-of-the-art results on QM9 and GEOM Drugs benchmarks, consistently outperforming strong baselines in novelty, atomic stability, and molecular stability.
- Computational Efficiency: FlexiFlow generates high-quality conformers at a fraction of the computational cost of traditional physics-based search tools like CREST.
- Task Flexibility: The framework demonstrates a successful transfer to protein-pocket conditioned generation, leveraging flexibility learned from other datasets even when the target dataset lacks multi-conformer data.


Weakness:
- Clarity on Reference Choice: The selection of the reference conformer x as the closest to the average conformation could be better justified against other theoretical alternatives, even if the authors note empirical results were indistinguishable.
- Protein Conditioning Evaluation: While the extension to protein conditioning is a strong conceptual point, the evaluation relies heavily on Autodock Vina scores. Vina is computationally cheap but less accurate than more rigorous binding free energy calculations.
- Scalability Limits: The appendix notes that GPU VRAM memory constraints limited the generation to 42 $y$ conformers simultaneously during the protein conditioning experiments, indicating potential scalability hurdles for massive ensembles.

---

> ### Author Rebuttal · Authors · 2026-03-30
>
> W1. The choice of reference conformer can, in principle, be application dependent and selected according to the specific task. We explored some reasonable options and did not find much impact: as noted, in our experiments using the minimum energy conformer as the reference, along with other reasonable alternatives, yields indistinguishable results (see Appendix E).
>
> W2. We agree with this comment. However, Vina is also widely used as a standard evaluation metric in the context of the previous protein-conditioned molecular generation studies, and as an open-source tool guarantees full reproducibility. More detailed binding energies estimates such as those from Free Energy Perturbation simulations, are more computationally expensive and further depend heavily on system setup, choice of the reference state, system preparation, how waters are handled - and these are out of scope for this study and cannot be readily scaled to a large and diverse test set. Instead, we prioritize reproducibility by the community as well as the comparison to the existing literature, although we will update the text to emphasize the limited expected accuracy of Vina scores.
>
> W3. Thanks, we missed to specify that it was a small consumer GPU. We have updated the manuscript to clarify that the experiments were conducted on an RTX 3060 GPU with 6 GB of VRAM. The observed limit of generating 42 conformers simultaneously is therefore a hardware-dependent constraint rather than a fundamental limitation of the method. Note that the current implementation is not fully optimized for memory efficiency and still produces around ~50 conformers on a "small consumer GPU". In a production-ready version, a key advantage of the approach is that the features of the input conformer ($x$) and the protein pocket can be computed once and reused, enabling the generation of a large number of output conformers ($y$) without incurring significant additional memory overhead. This design would substantially improve scalability for large ensemble generation.
>
> Q1. To be precise, each conformer represents a sample in phase-space and has an associated energy for a given potential energy function. We would expect a samples from a real system to be distributed according to Boltzmann distribution, and our ultimate aim when generating conformers is to be able to estimate thermodynamic observables of the system (binding energies, solubility etc). Therefore, generating high-energy conformations are not harmful because they do not contribute to observables (effectively wasted compute), but missing low-energy modes can lead to large errors in estimation. Molecular dynamics simulations are inefficient exactly because they generate highly auto-correlated samples, necessitating the use of metadynamics to more efficiently explore configuration space. In this context, a perfect conformer generator would sample directly and independently from the Boltzmann distribution, and correlation between samples is undesirable. Since we generate samples independently, there is no way for the model to know what is being sampled in parallel, which may conceivably lead to mode collapse or under-sampling of certain modes - however, we would argue that this is an artifact of incorrectly approximating the target distribution instead of explicitly lacking correlation between samples. Nonetheless, we observe in practice that conformers we sample do not exhibit mode collapse and provide both good recall and precision compared to expensive physics-based methods (see Figure 3). We have not looked for transition states and generally would not expect that these will be readily recovered considering the training data consists of essentially locally-minimized structures.
>
> Q2. Thanks for this question. In our framework, the atom and bond-type features are conditioned on the reference conformer ($x$). Specifically, the model uses the atomic identities and bond topology from $x$ as fixed inputs, and the generation of the target conformers ($y$) is conditioned on these features. As a result, variations in the initial noise used to generate $y$ conformers affect only the 3D coordinates, while the underlying 2D molecular graph (i.e., atom types and bond connectivity) remains unchanged. This design ensures that all generated conformers correspond to the same molecular graph at the structural level. However, we acknowledge that geometric deviations in the generated coordinates could, in principle, lead to physically implausible structures (e.g., distorted bond lengths or angles). Empirically, as reported in Appendix C.1, we do not observe significant energy outliers for smaller molecules, indicating stable and consistent conformations. For larger molecules, some energy outliers do occur (see Appendix C.2), which may reflect such geometric inconsistencies.

---

> > ### Author Rebuttal · Reviewer_JCph · 2026-04-04
> >
> > I thank the authors for resolving my issues, and I will maintain a positive rating.

---

> > > ### Author Response · Authors · 2026-04-05
> > >
> > > Thank you for the constructive discussion and comments. We really appreciate it. We are also happy to see the rebuttal was fully resolved.

---

### Official Review · Reviewer_2Mjt · 2026-03-12

**Soundness:** 2
**Presentation:** 3
**Significance:** 3
**Originality:** 3
**Overall Recommendation:** 4
**Confidence:** 4

**Summary:**

The paper introduces FlexiFlow, a method that extends flow matching to generate multiple conformations of a molecule at a time. The effectiveness of this design choice is evaluated on de-novo molecular generation, conformer generation and protein-conditioned ligand design tasks

**Compliance With Llm Reviewing Policy:**

Affirmed.

**Final Justification:**

The rebuttal addressed all of my questions and the quality of the paper did improve with the response.

**Key Questions For Authors:**

I have the following comments/questions for the authors:

1)There needs to be a controlled experiment where a model with (approximately) the same architecture is trained on the same dataset to generate one conformer at a time, in order to isolate the effect of multi-conformer training.

2)Why is the "reference" conformer necessary? An ablation study removing the reference conformer would help clarify its contribution to model performance.

3)For de novo molecular generation, the FlowMol3 work introduced several metrics that assess distribution mismatch between generated and training data distributions, \%PB-valid, OOD ring rate, FG-dev etc. The authors should report these values on their GEOM drugs benchmark.

4)Similarly, to evaluate conformer generation against baseline methods, the authors should conduct the standardized benchmark on the full SPICE/GEOM dataset, as done in Adjoint Sampling. This would provide a clearer indication of model performance and its standing relative to established benchmarks.

**Limitations:**

yes

**Strengths And Weaknesses:**

Strengths: The work introduces an interesting concept, proposing that training a model to generate multiple conformations simultaneously provides a richer learning signal than training on a single conformation alone. Competitive results across diverse benchmarks and tasks suggest this approach may prove broadly useful but requires further investigation and evidence.

Weaknesses: The method lacks an ablation study or controlled experiment to isolate whether multi-conformer generation genuinely drives the observed performance gains in molecular generators, as the improvements could alternatively be attributed to dataset expansion alone. De novo molecular design would also benefit from evaluation with additional metrics to more comprehensively assess performance relative to existing baselines. Additionally, standardized benchmarks are needed to rigorously evaluate the conformer generation task.

---

> ### Author Rebuttal · Authors · 2026-03-30
>
> W1. Thank you for your comment. We indeed observed that the multi-conformer training setup provides benefits beyond just dataset expansion. In Appendix C.7 we reported a plot showing that the multi-conformer generation approach transfer to the protein-conditioned ligand generation task, even when the dataset contains only static pockets without accompanying conformations. This suggests that the model is learning a more flexible representation of the conformational space of a molecule, which is not solely due to having more data from multiple conformers. We will clarify this point in the manuscript to better address your concern.
> We believe that our framework in unconditional molecular generation does not necessarily provide performance gains on 1-conformer generation metrics, but rather it allows us to do something that was not possible before, while designing the molecule generating the ensemble of conformers. This as reported in Section 5.2 (Fig.3) allow us to evaluate properties at the ensemble level, i.e. the coverage of low energy conformers, which is not possible with a single-conformer generator. We have also calculated the metrics suggest by the reviewer which are detailed in Q3.
>
> Q1. Great suggestion, thanks! We have trained SemlaFlow for 100 epochs on the full GEOM drugs dataset. The performances are reported in the Table below and are aligned to the one reported in Table 2 in the paper. We are also stressing the fact that the metrics are similar between SemlaFlow and FlexiFlow. We want to highlight that SemlaFlow model does not produce multiple conformers for a single structure, so it cannot be directly compared to the FlexiFlow output. However, we provide a practical approach to allow SemlaFlow to generate a multiple conformers, which further highlights why FlexiFlow is needed for handling multiple conformations.
> First, we generate a molecule and store the full trajectories of its categorical variables (atom types and bonds) during inference. We then rerun inference with new coordinate noise while fixing these categorical variables, yielding molecules with identical structures but different coordinates.
>
> We also tested a variant aimed at improving conformer energy by aligning new coordinate noise to the original noise using Earth Mover’s Distance. We generated 1,000 molecules with 11 conformers each ($x$ plus 10 $y$). While validity, stability, and novelty align with the original SemlaFlow results, the generated conformers exhibit extremely high energy states.
> | Model | Atom Stab ↑ | Mol Stab ↑ | Valid ↑ | E. (x) ↓ | S. E. (x) ↓ | E. (y) ↓ | S. E. (y) ↓ |
> |--------------------|-------------|------------|---------|----------|-------------|----------|-------------|
> | SemlaFlow | 99.9 | 99.9| 94.1    | 1.73   | 0.95 | 68558.20 | 68558.75    |
> | SemlaFlow + EMD| 99.9| 99.9   | 94.3    | 1.75| 0.92 | 4288.17  | 4287.36     |
> | FlexiFlow | 99.9| 99.9| 89.9    | 2.89     | 1.78 | 3.27     | 2.71        |
>
> Q2. Thank you for your question. The choice of the reference conformer is important for our method, as it serves as a conditioning variable that allows the model to learn the distribution of conformers. The reference conformer provides a fixed point of reference for the model to learn how to generate other conformers, given that connected atoms tend to preserve the same distances even when their conformations change. It is therefore not possible to ablate this component of our architecture while retaining the same training logic. Note also that, as above, SemlaFlow is a closely related model without the multi-conformer extension, and could serve as a reasonable surrogate for what is asked here. Finally, we stress again that FlexiFlow generates a different kind of object (an ensemble of conformers), and so comparisons about the conformer quality do not reflect the main focus of our contribution.
>
> Q3. While we agree that FlowMol3 (FM3) introduces novel metrics, it has not yet been peer-reviewed as of today. We nevertheless computed them following its setup (see table) and found results consistent with SemlaFlow. Our goal is not to improve single-conformer metrics, but to enable generation of conformer ensembles. Moreover, since FM3 is architecturally close to SemlaFlow, FlexiFlow could be adapted to FM3 to potentially retain its advantages.
> | Model             | %PDB Valid ↑ | FG dev ↓ | OOD Ring ↓ |
> |-----------------|-----------------|------------|----------|
> | FlowMol3 | 91.9±0.7| 0.27±0.03| 0.10±0.01|
> | FlexiFlow | 91.3±0.3| 0.38±0.02| 0.27±0.01|
> | SemlaFlow | 88.5±1.3| 0.35±0.02| 0.20±0.02|
> | EquiGAT-diff | 77.6±0.8| 0.58±0.03| 0.28±0.01|
>
> Q4. Thank you for your feedback. FlexiFlow is already trained on the full GEOM Drugs and we reported the same benchmarks for this dataset (see Cov/AMR in Table 2 of Adjoint sampling paper). Due to time constraints, we could not train on SPICE, but have reported results on QM9 which contains ~134k molecules.
>
> We will add all these extra insights to the Appendix

---

> > ### Author Rebuttal · Reviewer_2Mjt · 2026-04-03
> >
> > Thank you for addressing my questions. I believe the paper will be improved with the new results and I will increase my score to reflect this.

---

> > > ### Author Response · Authors · 2026-04-05
> > >
> > > Thank you for the constructive discussion and comments. We really appreciate it. We are also happy to see the rebuttal was fully resolved.

---

### Official Review · Reviewer_eh17 · 2026-03-13

**Soundness:** 4
**Presentation:** 3
**Significance:** 4
**Originality:** 3
**Overall Recommendation:** 5
**Confidence:** 3

**Summary:**

This paper targets a real bottleneck in 3D molecular generation: most flow/diffusion models generate one molecule with one conformation, while many downstream properties in drug discovery depend on the whole conformational ensemble rather than a single pose. FlexiFlow extends conditional flow matching so the model can jointly generate a molecular graph together with a reference conformation x and an arbitrary set of additional conformers.  Empirically, the paper reports very strong molecular-generation results on QM9 and GEOM Drugs

**Compliance With Llm Reviewing Policy:**

Affirmed.

**Key Questions For Authors:**

How critical is the conditional-independence assumption in practice?

Can the authors quantify how much is lost by factorizing the conformer ensemble into pairwise (x,y) terms, rather than modeling the full joint dependence among all conformers?


Why does GEOM Drugs validity drop relative to some baselines?

The paper is very strong on stability and novelty, but validity on GEOM Drugs is lower than several competitors. Is this a tradeoff caused by joint ensemble generation, by the architecture, or by training setup?


How does performance scale with the number of conformers requested at inference?

The paper states about 30% extra inference time to sample y, but how do quality and cost scale as the requested ensemble size grows?

**Limitations:**

yes

**Strengths And Weaknesses:**

Strengths:

1. The paper tackles an important and underexplored problem. The jump from generating a single 3D structure to generating a molecule plus a conformer set is conceptually meaningful for drug discovery, and the paper explains that motivation clearly

2. The main technical idea is clean. The decomposition from a joint flow on (x,S) into pairwise terms is simple, readable, and mathematically organized.

3. The empirical section is strong. Reported graph-generation metrics are excellent, and the conformer evaluation goes beyond standard validity/stability tables by including diversity, Cov, and AMR against CREST-based low-energy references.

Weakness:

1. The central modeling assumption is strong. The whole decomposition relies on conditional independence across conformers given the representative conformation, so the learned object is not obviously the full joint conformer ensemble in a rich correlated sense

2. Some metrics are not uniformly dominant. On GEOM Drugs, FlexiFlow is excellent on stability and novelty, but the reported validity is 92.0, which is below several baselines such as SemlaFlow, EQGAT-diff, FlowMol3, and Tabasco on that metric. So the method is not simply better in every direction.

---

> ### Author Rebuttal · Authors · 2026-03-30
>
> W1. Thanks for this comment. In our case, we believe that conditional independence is a fair assumption, as molecular conformers can be considered independent phase-space samples. This is a desiderata property because we want to explore phase-space in an uncorrelated manner. This is compared to running molecular dynamics, where the sampling is highly correlated. A potential downside is the risk of incomplete sampling of accessible conformers, if we generate many redundant conformers (i.e., mode collapse), but we have not observed this behavior empirically. See for e.g. Figure 3 (left) where we showed that the conformers are all different as their RMSD is >0 by a large margin.
>
> W2. Thank you for pointing this out. Indeed, the validity score of FlexiFlow on the GEOM Drugs dataset is slightly lower than some baselines. However, we want to emphasize that our method solves a more complex problem of modeling also the full distribution of conformers, which is a more challenging task, and the methods are not direct competitors since they solve different problems.
>
> Q1. In practice, the conditional-independence assumption allows us to factorize the joint distribution of conformers into pairwise ($x$,$y$) terms, which simplifies the modeling process and makes it computationally feasible.
>
> Q2. Most applications of conformational ensembles such thermodynamic integration treat conformers as independent samples (with Boltzmann weights), and there no natural reason to model conditionally dependence between conformers. Therefore, we feel this assumption in this context is reasonable. As mentioned before, the only potential issue we can think of, is that we cannot enforce exhaustive sampling of the conformational landscape, since each sample is independent of all others. However, we did not observe this in practice (see e.g. Figure 3).
>
> Q3-4. Great questions. The validity is slightly lower, but it is important to notice the we have trained the model only for 4 epochs. The reason for that is motivated by differences in dataset size. SemlaFlow was trained for 200 epochs on $\sim$250k samples from GEOM Drugs (as they considered only one conformer per molecule). In contrast, our setting considers all the conformers in GEOM Drugs which are $\sim$37M. This corresponds to a ratio of roughly 80$\times$ in dataset size, which is why we limited the training to four epochs in order to maintain a comparable number of parameters updates. That said, it is possible that training for more epochs would further improve the validity, but this is not the main focus of our work, which is to demonstrate that we can generate multiple conformers without affecting too much the quality of the generated molecules.
>
> Q5-6. The performance scales linearly with the number of conformers requested at inference. The extra 30% is due to a slightly more complex architecture which requires extra operations on $y$ compared to SemlaFlow. The quality of generated conformers remain consistent as the ensemble size increases, since each conformer is generated independently while conditioned on the same reference conformer $x$. We have reported in Appendix F the inference time cost comparison per conformer, and we are actively working on optimizing the implementation to further improve efficiency for larger ensemble generation.

---

> > ### Author Rebuttal · Reviewer_eh17 · 2026-04-05
> >
> > Thanks for the detailed rebuttal. The reply about the loss of validity is clear and I shall keep the positive score.

---

> > > ### Author Response · Authors · 2026-04-06
> > >
> > > Thank you for the constructive discussion and comments. We really appreciate it. We are also happy to see the rebuttal was fully resolved.

---

### Decision · Program_Chairs · 2026-04-30

**Decision:**

Accept (regular)

**Comment:**

The paper proposes an extension of molecular generative flow-matching models that allows for sampling molecules along with multiple conformations, achieving similar ensemble coverage to state-of-the-art physics-based methods.

The reviewers provided detailed and helpful comments and largely recognize the originality and usefulness of the approach. Extending flow-matching to generating distributions represents a technically interesting topic. Though, there are some points which would make the paper stronger. In particular, the data might not provide the necessary diversity to fully evaluate the capabilities of the approach. Furthermore, the existing adjoint sampling seems to be to be working similarly well; the authors show performance differences but those are small.